# Substitution of polysorbates by plant-based emulsifiers: impact on vitamin D bioavailability and gut health in mice
Ángela Bravo-Núñez [1,2] ✉, Angélique Berthomé[1], Charlotte Sabran[1], Donato Vairo [1], Jean-Charles Martin[1], Katherine Alvarado-Ramos[1], Benoit Chassaing[3], Julie Tomas [4] & Emmanuelle Reboul [1] ✉

Although long considered safe, recent data have shown that emulsifiers such as polysorbates promoted intestinal inflammation and were associated with increased risks of developing chronic pathologies. We evaluated the potential of plant-based emulsifiers (pea protein isolate, PPI, and corn arabinoxylans, CAX) as alternatives to Polysorbate 80 (Tween 80, T80). Combining PPI and CAX led to a similar vitamin $D_3$ bioavailability to T80 in vitro and in vivo in mice. We then exposed female and male mice to dietary doses of emulsifiers in oil-in-water emulsions (180 mg/kg/day for T80, 5 days/week) for 11 weeks. Conversely to previous studies conducted with higher doses of emulsifiers, T80, PPI, and PPI + CAX groups were similar to the control group (oil alone) in terms of physiological characteristics and inflammation biomarkers. However, LPS-specific serum IgG levels were reduced in the PPI ($-31.05\%$, $p = 0.0006$) and PPI + CAX ($-34.66\%$, $p = 0.0001$) groups compared to the T80 group at the end of the intervention. Exposure to T80, but not to PPI or PPI + CAX, reduced the distance between bacteria and the jejunal epithelium ($-60.67\%$, $p = 0.0779$) and significantly increased Firmicutes_D phylla in male mice. Overall, we showed that a combination of pea protein and arabinoxylans appears as a sustainable alternative to polysorbates for vitamin $D_3$ delivery.

Micronutrient deficiency, a widespread form of malnutrition, poses significant challenges to human health and development worldwide. Despite its crucial role in phosphocalcic metabolism[1] and immunity[2], vitamin D deficiency is a global health issue affecting all age groups. Strategies to tackle this deficiency include the use of supplements or foods containing emulsions enriched in vitamin D. These emulsions are often stabilized with synthetic emulsifiers (see for review[3,4]). These emulsifiers have long been considered safe, indigestible, and mainly excreted in feces[5,6]. However, a growing body of evidence has now highlighted their detrimental effect on health[7–13], as recently reviewed by Whelan et al.[14] in detail. The deleterious effects of synthetic emulsifiers seem to be largely mediated by an alteration of the intestinal microbiota composition and distribution[7,9,11,13]. However, these studies addressed the impact of high concentrations of emulsifiers, far above of plausible exposure levels. Although high dose toxicity testing is essential to define thresholds for adverse effects and clarify potential risks associated with chronic or accidental overconsumption, it does not represent actual exposure. Indeed, a concentration of 1% in water, used in the above-cited studies, results

in an exposure of 1800–3600 mg/kg animal/day, i.e., ~146–292 mg/kg/day when converted to human dose[15], while the acceptable daily intake proposed by the European Food Safety Authority (EFSA) is of 25 mg/kg/day for T80[16]. Although consumers likely surpass EFSA levels, it is unlikely that they will reach such high concentrations. Nevertheless, the results of these studies are supported by data obtained using the in vitro SHIME model with lower concentrations of synthetic emulsifiers[8], by epidemiological data[17,18], as well as by a recent feeding study in healthy volunteers[12].

Existing alternatives to synthetic emulsifiers include milk proteins, lecithin (mostly from soy), or saponins[19–22]. However, emulsifiers from animal origin or unsustainable crops may not constitute long-term solutions, considering the need for a transition towards more sustainable foods and ingredients[23]. In this line, proteins from legumes and arabinoxylans from corn by-products, which can be found in significant amounts in regular diets, display a high potential as emulsifiers[24,25] and for fat-soluble vitamin delivery[26], with potential health benefits associated with corn arabinoxylans[27–29]. In a previous study, we reported that emulsions stabilized

[1]Aix Marseille Univ, INRAE, INSERM, C2VN, Marseille, France. [2]University of Valladolid, Valladolid, Spain. [3]Microbiome-Host Interactions, Institut Pasteur, Université Paris Cité INSERM U1306, Paris, France. [4]Aix Marseille Univ, CNRS, INSERM, CIML, Marseille, France. ✉e-mail: angela.bravo-nunez@univ-amu.fr; angela.bravo@uva.es; Emmanuelle.Reboul@univ-amu.fr

with legume proteins and arabinoxylans presented similar physicochemical characteristics in terms of droplet size and stability than emulsions stabilized with synthetic emulsifiers such as polysorbates (Tween 80)[26]. In this context, the objective of this study was thus to explore the potential of pea proteins and arabinoxylans as alternatives to synthetic emulsifiers to deliver vitamin $D_3$ in vitro and in mice, and to evaluate their impact on health after chronic consumption at dietary doses in both female and male mice.

## Results

### Emulsions stabilized with pea proteins and corn arabinoxylans are as effective as a Tween 80 emulsion to promote in vitro vitamin $D_3$ bioaccessibility, but not uptake by cultured intestinal cells

The first part of the study aimed at exploring the potential of legume proteins, i.e., pea protein isolates (PPI) or lentil protein isolates (LPI), and corn arabinoxylans (CAX) as alternatives to synthetic emulsifiers to deliver dietary vitamin $D_3$. Tested emulsion compositions are presented in Table 1. In vitro digestion revealed that vitamin $D_3$ from emulsions stabilized with PPI and PPI + CAX (0.15% or 0.9%) had a bioaccessibility equivalent to that of vitamin $D_3$ from Tween 80 (T80) emulsion (Fig. 1A), meaning that similar amounts of vitamin $D_3$ were potentially absorbable at the end of the in vitro digestion process. 1% CAX emulsion was the only condition that significantly reduced vitamin $D_3$ bioaccessibility compared to the control emulsifier T80 (p = 0.002).

As emulsions stabilized with PPI and PPI + CAX (at both CAX levels) allowed the highest vitamin $D_3$ bioaccessibility, they were chosen for further analyses using Caco-2 cells, and LPI and LPI + CAX were not further analyzed. Caco-2 cells were exposed to the diluted micellar phase (1/8) of the in vitro digestions. When needed, to allow an accurate quantification, diluted micelles were enriched with vitamin $D_3$ to reach a concentration of 0.23 μM. Vitamin $D_3$ uptake by highly differentiated Caco-2 cells (Fig. 1B) showed that vitamin uptake was significantly higher when using T80 than when using either PPI or PPI + CAX at both CAX levels (p < 0.0001).

### Emulsions stabilized with pea proteins and arabinoxylans are as efficient as a Tween 80 emulsion to promote vitamin $D_3$ absorption in mice

To validate in vivo our in vitro results, we force-fed mice two doses of vitamin $D_3$ in emulsions stabilized with either T80 or PPI + 0.9%CAX (from now on referred as PPI + CAX, as PPI + 0.15%CAX was not further tested to reduce the number of animals). Plasma triglycerides (TG) and vitamin $D_3$ concentrations were reported at baseline, and during a 6h-postprandial period following force-feeding with oil-in-water emulsions enriched in vitamin $D_3$ (Fig. S1).

Plasma TG postprandial response (Fig. 1C, F) and TG accumulation in the small intestine and liver (Fig. 1D, E, G, H) were neither affected by emulsifiers nor by vitamin $D_3$ concentration.

Postprandial vitamin $D_3$ concentrations (Fig. 1I, L) remained similar when using either T80 or PPI + CAX for both vitamin $D_3$ tested concentrations (25 or 250 μg/mouse). They were only significantly different at t = 6 h with the lowest dose of vitamin $D_3$ (1.38 ± 0.11 μmol/L for T80 vs

1.09 ± 0.07 μmol/L for PPI + CAX, p = 0.027, Fig. 1I), also impacting vitamin $D_3$ postprandial response when expressed as AUC (6.39 ± 0.15 for T80 vs 5.44 ± 0.22 for PPI + CAX, p = 0.016). Both emulsifiers led to a similar profile of vitamin $D_3$ accumulation along the duodenal-ileal axis of the intestine (Fig. 1J, M), but not in the ileum with 250 μg of vitamin $D_3$ (334.07 ± 39.33 mmol/g for T80 vs 208.60 ± 32.39 mmol/g for PPI + CAX, p = 0.018, Fig. 1M). The emulsifiers did not impact liver content in vitamin $D_3$ (Fig. 1K, N).

### Chronic exposure to emulsifiers did not affect mouse total weight, organ weights, intestinal permeability and vitamin status

We next assessed the effect of a chronic exposure to emulsifiers through an intervention of 11 weeks (representing 10–15% of mouse lifespan, equivalent to ~10 years of consumption in humans) during which mice received dietary doses of either T80 (180 mg/kg mice/day), PPI (360 mg/kg mice/day), and PPI + CAX (441 mg/kg mice/day) 5 days a week. These doses represent plausible dietary doses (see Materials and Methods section and Supplementary Note 1 for detailed justification). Control mice received non-emulsified olive oil (i.e., 20 μL) to achieve equivalent oil intake (Fig. S2).

Emulsifiers did not significantly affect mouse food intake (Fig. S3A, B), total weight gain (Fig. S3C, D), spleen weight (Fig. 2B) or small intestine length (Fig. 2C). Liver weight (Fig. 2A) and large intestine length (Fig. 2D) were sex-dependent (higher values for males, p < 0.001 and p = 0.0062, respectively), but values were not affected by treatment for each sex.

Intestinal permeability (Fig. 2E) remained unchanged after 11 weeks of exposition.

At the end of the 11 week-intervention, there was no significant difference regarding plasma fat-soluble vitamin concentrations (vitamin $D_3$ in the form of 25-hydroxyvitamin $D_3$, i.e., 25(OH)$D_3$; vitamin E in the form of α-tocopherol; and vitamin A in the form of retinol) between treatments in both male and female mice (Fig. 2F–H). However, vitamin plasma concentrations were sex-dependent (p = 0.0046, 0.0148, and <0.0001 for vitamin $D_3$, E, and A, respectively). No interaction between sex and treatment was observed.

### Chronic exposure to emulsions stabilized with pea proteins and arabinoxylans improved some inflammatory biomarkers compared to a Tween 80 emulsion and olive oil alone

We then examined the effect of emulsifiers at dietary levels on different inflammatory biomarkers in mouse plasma and feces at selected time points.

After 3 weeks of emulsifier treatment, LPS-specific serum IgG level was significantly decreased in the PPI + CAX group compared to the control group (oil alone) (p = 0.0352, Fig. 3A). This decrease was observed throughout the intervention (p-values of 0.0512, 0.0111, and 0.0357 at weeks 6, 9, and 11, respectively). A decreased response was also observed in the PPI and T80 groups at week 9 compared to the control group (p = 0.0018 and 0.0236, respectively). Finally, T80 exposure generated a higher LPS-specific serum IgG response than PPI + CAX exposure at weeks 3 (p = 0.0074) and 11 (0.0001), and than PPI exposure at week 11 (p = 0.0006) (Fig. 3A).

The level of Flagellin FliC-specific serum IgG was higher at week 6 for the PPI + CAX group than for the control (p = 0.0423), T80 (p = 0.0011)

**Table 1 | Detailed emulsion formulation (mg/g emulsion)**

| | Tween 80 | Pea protein isolate | Lentil protein isolate | Corn arabinoxylans | MiliQ Water | Olive oil[a] |
|---|---|---|---|---|---|---|
| T80 | 18 | - | - | - | 882 | 100 |
| PPI | - | 36 | - | - | 864 | 100 |
| PPI + 0.9% CAX[b] | - | 36 | - | 8.1 | 855.9 | 100 |
| PPI + 0.15% CAX | - | 36 | - | 1.35 | 865.65 | 100 |
| LPI | - | - | 36 | - | 864 | 100 |
| LPI + 0.9% CAX | - | - | 36 | 8.1 | 855.9 | 100 |
| LPI + 0.15% CAX | - | - | 36 | 1.35 | 862.65 | 100 |

[a]Oil was loaded with vitamin $D_3$ for bioaccessibility and bioavailability experiments at low and high doses to achieve the vitamin $D_3$ concentration in the emulsions of 0.975 and 216.54 or 2165.66 μM, respectively. For cell uptake, diluted micelles (1/8) from bioaccessibility experiments were spiked with vitamin $D_3$ to achieve a final concentration of 0.23 μM.
[b]Also referred as PPI + CAX in the text.

**Fig. 1 | Vitamin D in vitro bioaccessibility and uptake, and in vivo bioavailability in mice.** 2% Tween 80 group (T80, circles), 1% Corn arabinoxylans (CAX, circles), 4% Pea protein isolate (PPI, squares), 4%PPI + 0.15%CAX (PPI + 0.15%CAX, diamonds), 4%PPI + 0.9%CAX (PPI + 0.9%CAX or PPI + CAX, triangles) 4%LPI (LPI, circles), 4% LPI + 0.15%CAX (LPI + 0.15CAX, circles), 4% LPI + 0.9%CAX (LPI + 0.9CAX, circles). **A** Vitamin $D_3$ bioaccessibility assessed by in vitro digestion. **B** Vitamin $D_3$ uptake by highly differentiated Caco-2 TC7 cell monolayers. Vitamin D concentration= 0.23 μM. C-E and I-K panels correspond to in vivo experiments in which mice were force-fed emulsions enriched with 25 μg of vitamin $D_3$. **F–H, L–N** panels correspond to in vivo experiments in which mice were forced-fed emulsions enriched with 250 μg of vitamin $D_3$. **C, F** Postprandial plasma triglyceride concentration. **D, G** Intestinal triglyceride content. **E, H** Liver triglyceride content. **I, L** Postprandial plasma vitamin $D_3$ concentration. Area Under the Curve (AUC). **J, M** Intestinal vitamin $D_3$ content. **K, N** Liver vitamin $D_3$ content. For **A**, different letters mean significant differences between samples. For other panels, an asterisk indicates significant differences compared to T80 group (*, $p < 0.05$; **, $p < 0.01$; ***, $p < 0.001$, ****, $p < 0.0001$). For in vivo experiments, all graphic values are represented as mean ± SEM with n = 6 male mice/group.

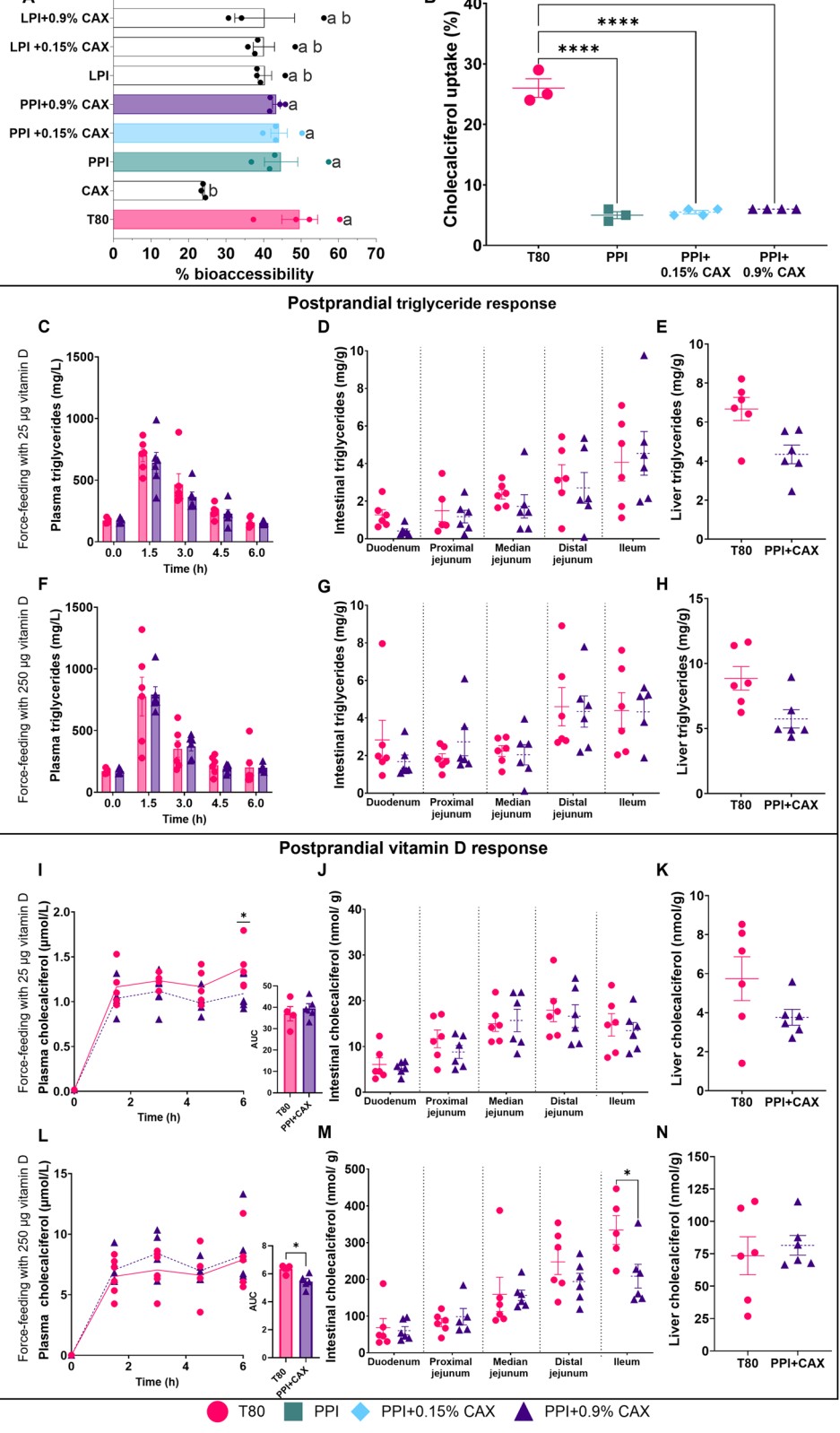

and PPI ($p = 0.0011$) groups, but this increase was only transient (Fig. 3B). On the contrary, at week 11, PPI treatment showed a lower Flagellin FliC-specific serum IgG response than the control group ($p = 0.049$).

Most measured pro-inflammatory cytokines (IL-23, IFN-γ, TNF-α, IL-12p70, IL-1β, IL-6, IL-17A, & GM-CSF) were not upregulated in emulsifier groups compared to the control group (Fig. S4A, C, D, F, G, H, I, J). For IL-1α, PPI and PPI + CAX groups showed a higher response compared to the

control group ($p = 0.0027$ and $p = 0.0059$, respectively), and the PPI group showed a higher response than the T80 group ($p = 0.0111$ at week 9). A transient MCP-1 higher response was observed at weeks 3 and 6 in the PPI + CAX group compared to the control group ($p = 0.0426$ and 0.0194, respectively). In parallel, the anti-inflammatory cytokines IL-10 and IL-27 were significantly increased in the presence of PPI + CAX. IL-10 response was increased in the presence of PPI + CAX compared to control at weeks 3

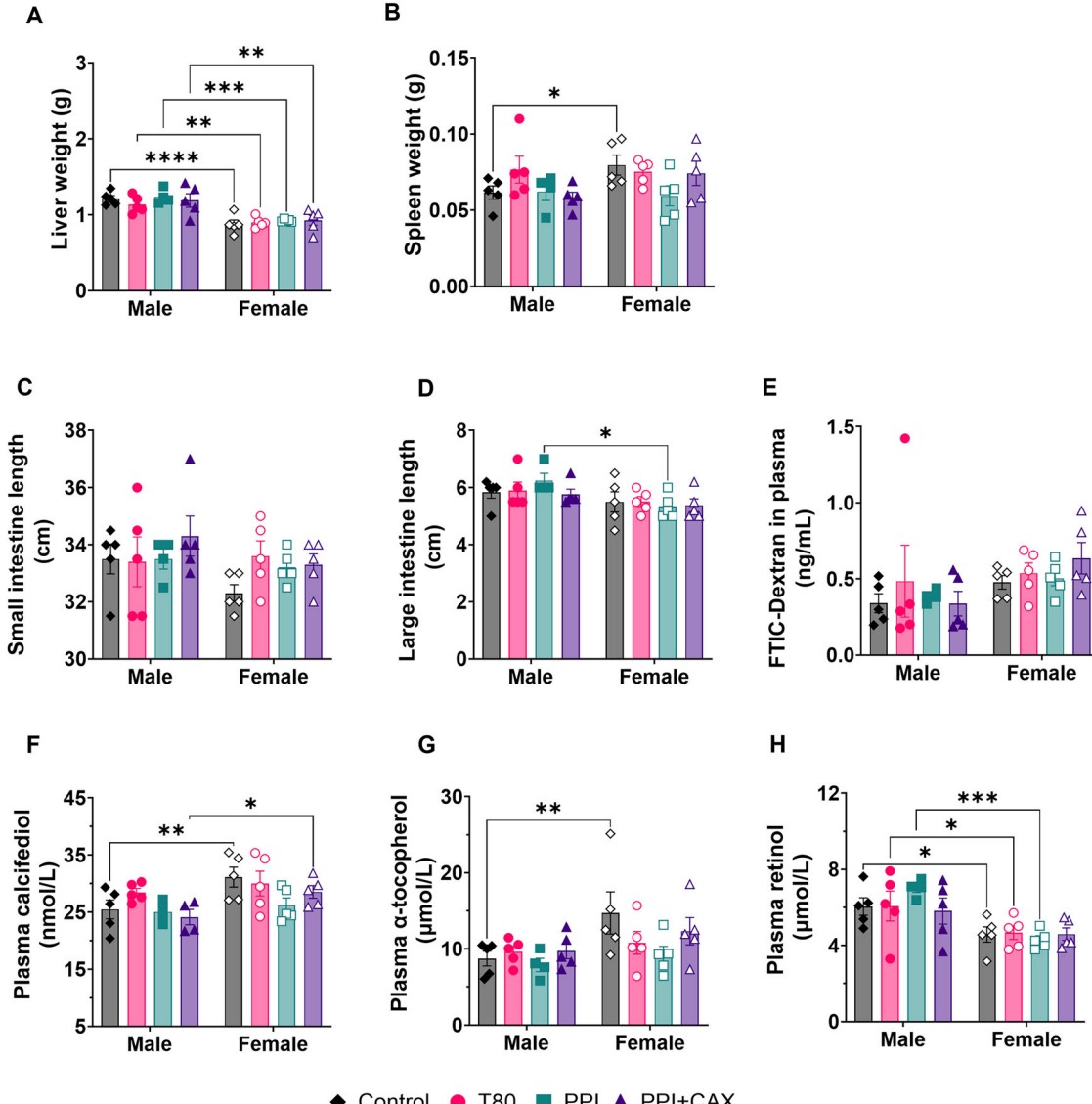

Control ◆   T80 ●   PPI ■   PPI+CAX ▲

**Fig. 2 | Physiological parameters and vitamin status of mice after chronic exposure to emulsifier-stabilized emulsions at dietary doses considering sex.** The olive-oil group (Control, diamonds), 2% Tween 80 group (T80, circles), 4% Pea protein isolate (PPI, squares), 4%PPI + 0.9% Corn arabinoxylans (PPI + CAX, triangles) were exposed to dietary doses of emulsifiers 5 days a week for 11 weeks. Mice were euthanized at fast. Males are represented by full symbols and females by empty symbols. **A** Liver weight, **B** Spleen weight, **C** Small intestine length, **D** Large intestine length, **E**) Plasma FITC-dextran concentration, **F** plasma 25(OH)vitamin D₃, **G** plasma α-tocopherol, and **H** plasma retinol. An asterisk indicates significant differences between groups (* $p < 0.05$; ** $p < 0.01$; *** $p < 0.001$). All graphic values are represented as mean ± SEM with n = 10 mice/group including 5 males and 5 females.

and 6 ($p = 0.0296$ and $0.0499$, respectively) (Fig. S4K). IL-27 response was also increased by PPI + CAX compared to the T80 group at weeks 6 and 9 ($p = 0.0175$ and $0.029$, respectively) (Fig. S4L). Finally, IFN-β expression, with both pro- and anti-inflammatory effects, was not affected by emulsifiers (Fig. S4M).

We also explored selected key biomarkers of fecal inflammation (flagellin, lipocalin-2, and LPS). Flagellin levels were higher in T80 group than in PPI ($p = 0.001$) and PPI + CAX ($p = 0.015$) groups at week 3, and higher in PPI + CAX group than in all other groups at week 9 ($p = 0.0057$, <0.0001, and 0.0007 for control, T80, and PPI, respectively) (Fig. 3C). When viewed as a whole, all the emulsifier groups tended to decrease these 3 fecal biomarkers over the intervention compared to the control group (Fig. 3C–E).

**Tween 80, but not pea proteins and arabinoxylans, modifies the microbiota spatial localization in the upper gastrointestinal tract**

We then evaluated the impact of emulsifiers on the colon and jejunum in terms of morphology, mucus, and bacteria distribution to understand whether the differences observed for inflammation could be related to changes in the intestine.

No effect on morphology and microbial localization was observed in the colon, with the bacteria remaining at distance from the inner mucus layer (Fig. S5).

Emulsifiers did not significantly affect the jejunum intestinal morphology when not considering sex (Figs. 4A–C and S6 for hematoxylin-eosin pictures). When taking sex into consideration, male mice exposed to PPI had longer villus length than male mice exposed to PPI + CAX ($p = 0.0471$, Fig. 4A), while males exposed to T80 had wider villus than males exposed to PPI ($p = 0.0104$, Fig. 4B). Importantly, bacteria tended to be closer to the epithelium for T80-treated mice compared to the control group ($p = 0.0779$, Fig. 4D, E). When analyzing the distance by sex, the same tendency was maintained for males ($p = 0.0822$) but not for females ($p = 0.1472$) (Fig. 4E).

To assess whether the proximity of bacteria to the intestinal epithelium was connected to changes in mucus layer, we next stained the jejunum

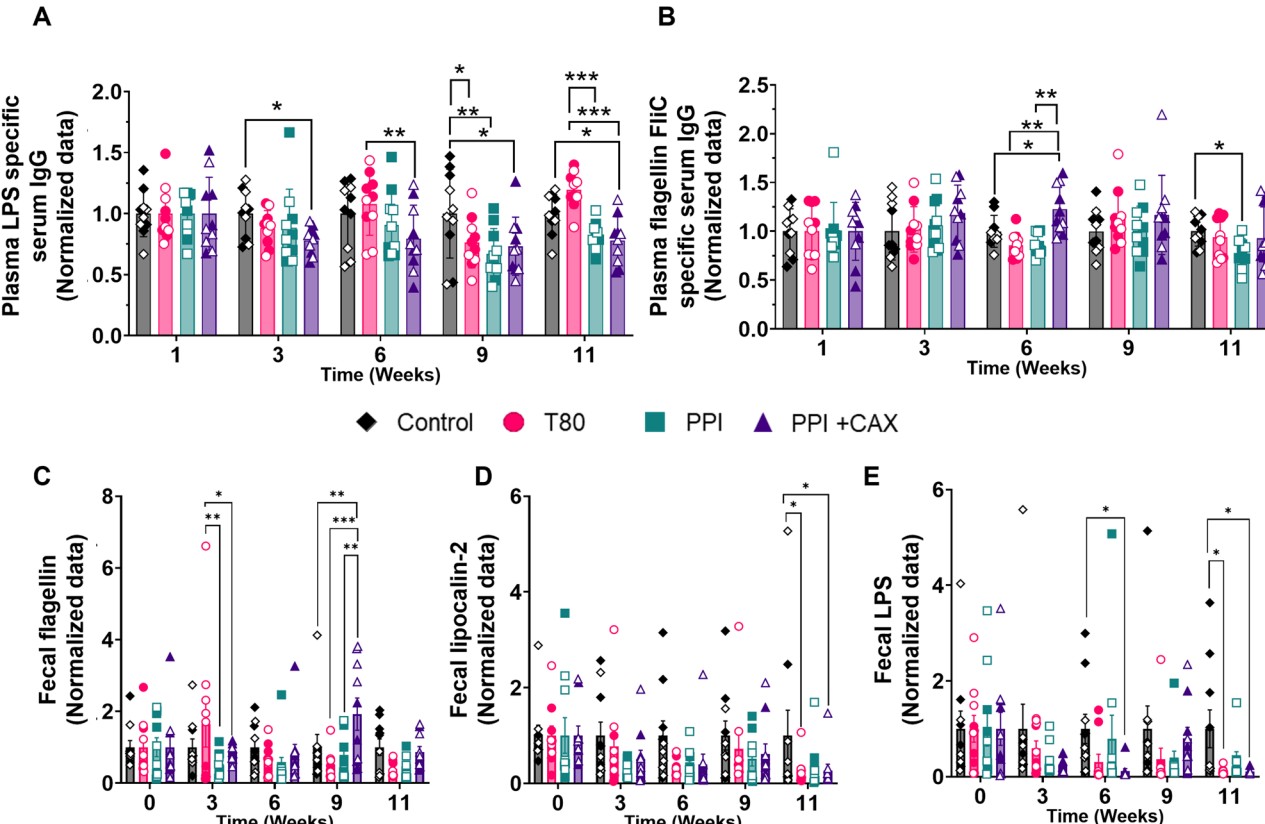

**Fig. 3 | Plasma and fecal biomarkers of inflammation after chronic exposure of mice to emulsifier-stabilized emulsions at dietary doses.** The olive-oil group (Control, diamonds), 2% Tween 80 group (T80, circles), 4% Pea protein isolate (PPI, squares), 4%PPI + 0.9% Corn arabinoxylans (PPI + CAX, triangles) were exposed to dietary doses of emulsifiers 5 days a week for 11 weeks. Feces were collected in ARN-free tubes. Mice were euthanized at fast. Males are represented by full symbols and females by empty symbols. Relative values of plasma concentrations of specific IgG against **A** LPS and **B** Flagellin FliC. Normalized data of fecal concentrations of **C** Flagellin, **D** Lipocalin-2, and **E** LPS. Relative values were normalized to the control group, including both males and females. An asterisk indicates significant differences between groups (*p < 0.05; **p < 0.01; ***p < 0.001). All graphic values are represented as mean ± SEM with n = 10 mice/group, including 5 males and 5 females.

for Mucin 2, the major glycoprotein of the intestinal mucus layer (MUC2, magenta) and the Vicia Villosa Lectin (VVA, cyan). For male mice, the mucus layer in the T80 group looked thicker and more adherent to the epithelium than in the other groups and mainly stained with the VVA antibody (Figs. 4D and S7). This effect was not observed in females. Regardless of the sex, secretion of mucus seemed to be altered in the presence of T80: Fig. 4D showed goblet cells full of mucus, which was not observed for other groups. In addition, the amount of both MUC2 and VVA in the lumen seemed limited for T80 in comparison to the other emulsifiers (Fig. S7).

To confirm whether the changes observed in the mucus layer could explain the proximity of bacteria to the epithelium observed in the T80 condition, we analyzed the expression of genes involved in i) mucus production and release (*Muc2, Muc3, Klf4* and *Meprin-β*) and ii) antimicrobial peptide *Reg3-γ* in the different parts of the jejunum. Overall, no significant difference was observed between the different groups (Fig. S8), but when looking for sex-specific responses, *Muc3* expression in females and *Meprin-β* expression in males were modulated by treatments (Fig. 5A, C, E). In females exposed to T80, *Muc3* was overexpressed in the proximal jejunum compared to PPI or PPI + CAX groups (p = 0.0148 and 0.0120, respectively). *Muc3* was also overexpressed in the distal jejunum of females exposed to PPI compared to females exposed to T80 and PPI + CAX (p = 0.0184 and 0.0371, respectively), (Fig. 5A, C). Finally, the expression of *Muc2* and *Muc3* in the proximal jejunum was upregulated for females compared to males in the T80 group (p = 0.033 and 0.0051 for *Muc2* and *Muc3*, respectively), while the expression of *Muc2* and *Muc3* was downregulated in the median jejunum of females compared to males in the PPI

group (p = 0.0214 and 0.0103, respectively). (Fig. 5A, C). Male mice exposed PPI + CAX displayed an upregulated expression of *Meprin-β* in the distal jejunum compared to the control and T80 groups (p = 0.0313 and 0.0179, respectively) (Fig. 5E). Overall, these data showed that visual differences between males and females exposed for T80 were linked to sex-dependent expression of *Muc2* and *Muc3* in the proximal jejunum.

No differences were observed for the antimicrobial peptide *Reg3-γ* when comparing groups (Fig. S8), but *Reg3-γ* expression was higher in females than in males exposed to T80 in the median jejunum (p = 0.005, Fig. 5D). When analyzing sex separately, *Reg3-γ* was overexpressed in the median jejunum of females of the T80 group compared to the PPI group (p = 0.0348).

We finally assessed whether the bacteria spatial localization could influence the expression of genes related to inflammation in the jejunum. The inflammation signaling genes *TNF-α* and *CXCL-1* were not affected by emulsifiers. *IL-6* expression was downregulated by T80 treatment in the median jejunum compared to the control condition (p = 0.0281) (Fig. S8). Sex-dependent analyses showed that this difference was driven by the significantly higher *IL6* expression in the control group than in the T80 or PPI female groups (p = 0.0016 and 0.0414, respectively, Fig. S8). However, *IL-6* expression was not measurable in all individuals, or the number of Cp needed for detection was high, meaning low expression in all samples. Therefore, these significant differences may be misleading.

**Emulsifiers modify microbiota composition at the phylum level**

To further investigate the effect of emulsifiers at dietary doses on the microbiota, we analyzed fecal microbiota composition at week 11 by 16S rRNA gene sequencing. Fecal α-diversity indexes (Shannon, Simpson, and

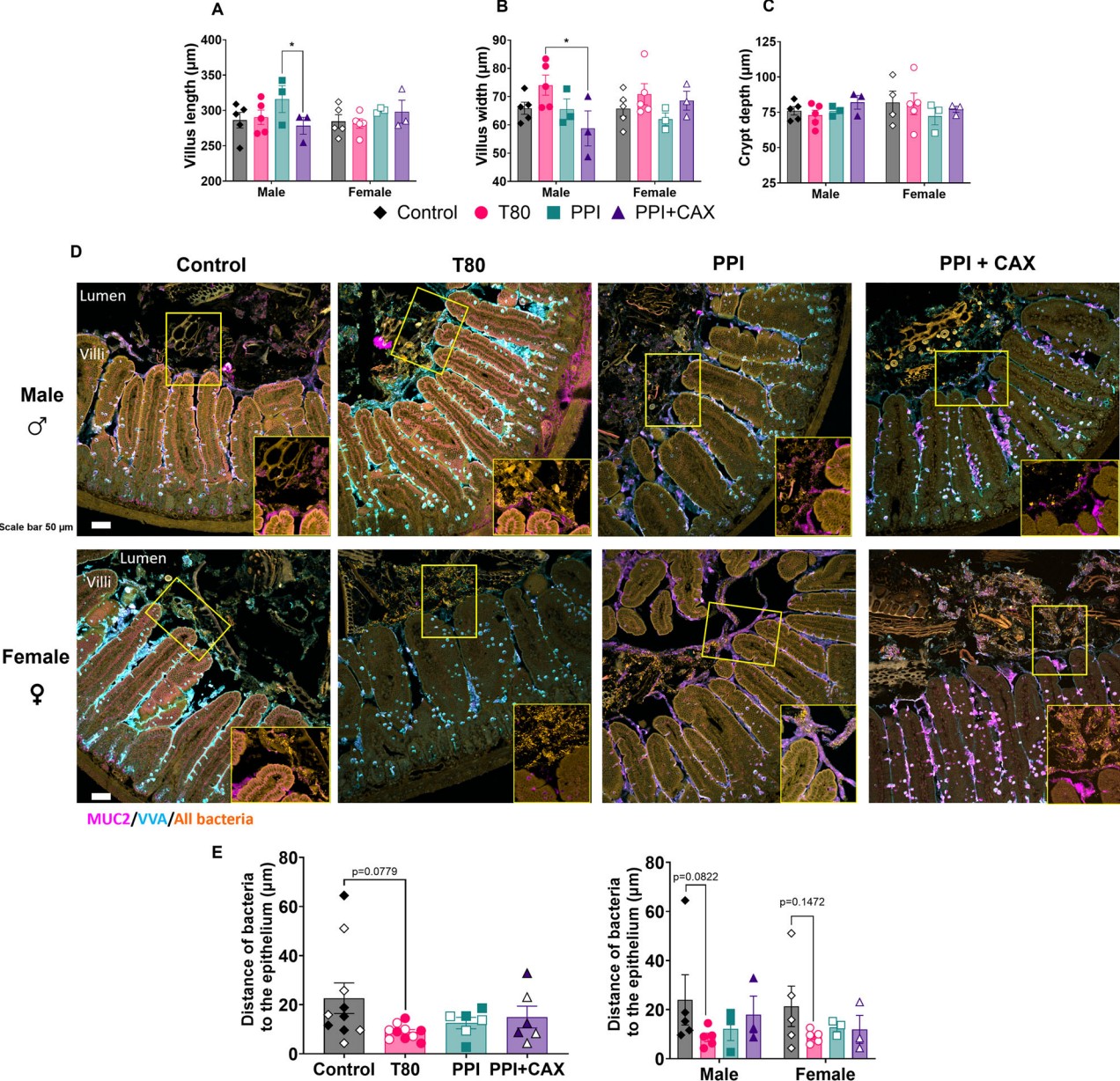

**Fig. 4 | Jejunal characteristics and microbiota distance to the jejunal epithelium after chronic exposure of mice to emulsifier-stabilized emulsions at dietary doses.** The olive-oil group (Control, diamonds), 2% Tween 80 group (T80, circles), 4% Pea protein isolate (PPI, squares), 4%PPI + 0.9% Corn arabinoxylans (PPI + CAX, triangles) were exposed to dietary doses of emulsifiers 5 days a week for 11 weeks. Mice were euthanized at fast. Males are represented by full symbols and females by empty symbols. Proximal-median jejunal tissues were recovered for **A** Villus length, **B** Villus width and **C** Crypt depth measures from hematoxylin-eosin staining (See Fig. S6); and **D** Representative spectral confocal imaging projections of median jejunum from mice stained by Fluorescence in situ Hybridization (FISH) for all bacteria (Eub-338 probe, orange), Mucin 2 (MUC2, magenta) and Vicia Villosa Agglutinin (VVA, cyan). Bars, 50 μm. **E** Quantification of the distance between the microbiota (Eub-338 probe, orange) and the intestinal epithelium (top of villi) using ZEISS ZEN 3.7 software line tool (Carl Zeiss Microscopy). All graphic values are represented as mean ± SEM for n = 6–10 mice/group including 3–5 males and 3–5 females, with at least 2–5 measures/mice. An asterisk indicates significant differences between groups (* p < 0.05; ** p < 0.01).

Evenness) showed no significant differences between groups when not considering sex (Fig. 6A–C).

Female mice showed significantly lower values than male mice for all diversity indexes (Fig. 6A–C). Males and females also clustered differently at the fecal bacterial β-diversity level, with the Bray Curtis or Jaccard analyses showing that individuals from groups exposed to any of the emulsifiers clustered together at a greater distance from the control group (Fig. 6D).

Firmicutes_D, Bacteroidota, and Firmicutes_A were the most abundant phyla for all mice (Figs. 6E and S9). Firmicutes_D relative abundance was higher in the T80 group than in the control group (p = 0.0091) in male mice, but not in females.

A sexual dysmorphism was observed when analyzing treatment effect between sex groups. For the control and PPI + CAX groups, female mice showed a higher Firmicutes_D relative abundance than males (p = 0.002 and 0.0049 for control and PPI + CAX groups, respectively), while male mice showed a higher Firmicutes_A relative abundance than females (p = 0.0059 and 0.0006 for control and PPI + CAX, respectively).

Bacteroidota relative abundance was lower in the PPI group than in the control group in male mice (p = 0.0375).

No differences were observed for Actinobacteriota, while significant differences were observed for Verrucomicrobiota between males and females for the control group (p = 0.0209).

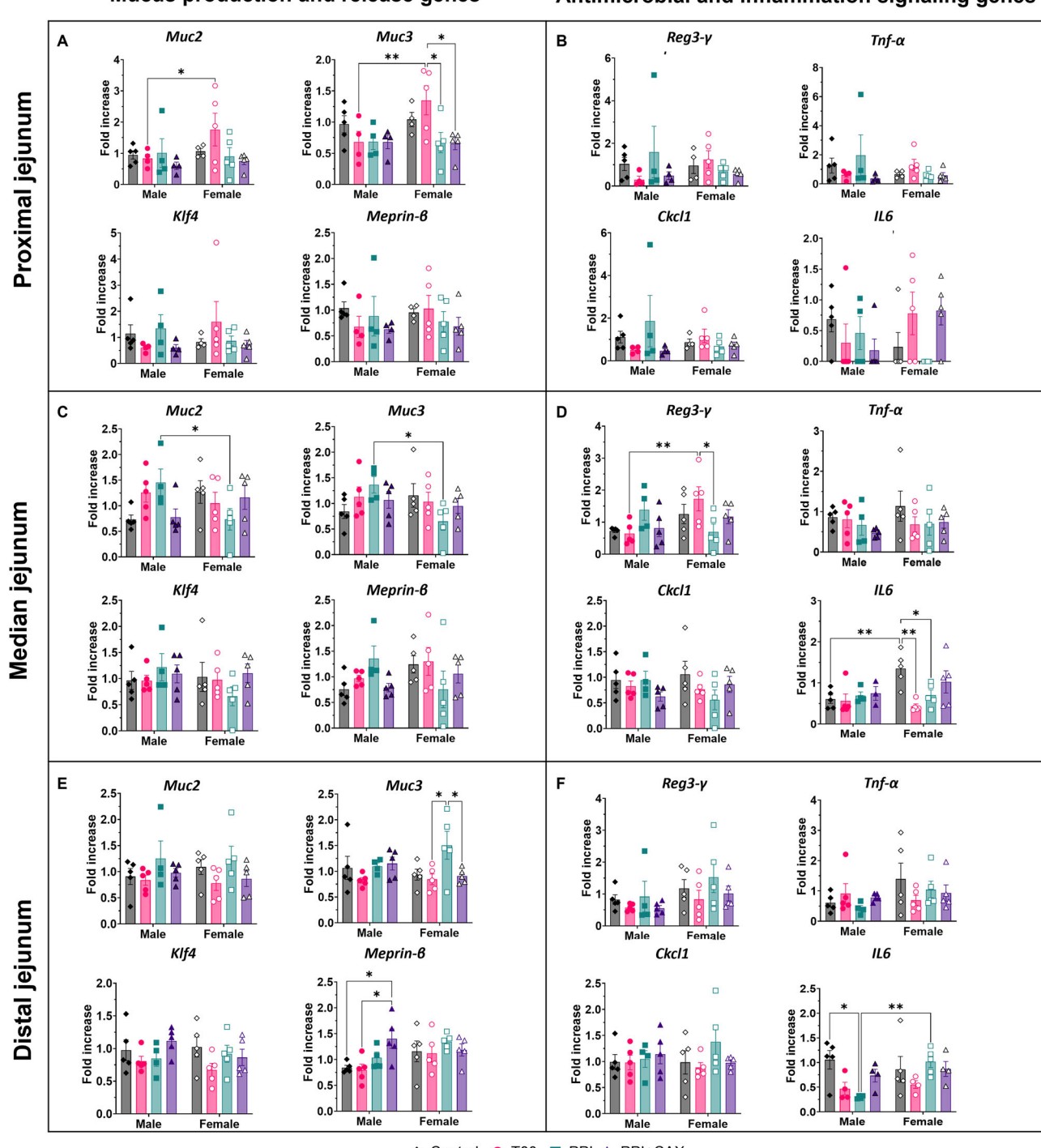

**Fig. 5 | Intestinal gene expression analyses related to mucosal defense systems after chronic exposure of mice to emulsifier-stabilized emulsions at dietary doses by sex.** The olive-oil group (Control, diamonds), 2% Tween 80 group (T80, circles), 4% Pea protein isolate (PPI, squares), 4%PPI + 0.9% Corn arabinoxylans (PPI + CAX, triangles). Males are represented by full symbols and females by empty symbols. Relative values of mucus production and release genes (**A** proximal jejunum, **C** median jejunum, and **E** distal jejunum), and antimicrobial and inflammation signaling genes (**B** proximal jejunum, **D** median jejunum, and **F** distal jejunum). All graphic values are expressed as mean ± SEM of relative values normalized to the control group including both males and females, with n = 10 mice/group including 5 males and 5 females. An asterisk indicates significant group differences (*p < 0.05; **p < 0.01).

For Desulfobacterota and Firmicutes_B, a clear effect of all emulsifiers was observed in male mice, as their relative abundance was significantly decreased compared to the control group (Desulfobacterota: p = 0.0002, 0.0002, and 0.0001 for T80, PPI, and PPI + CAX, respectively;

Firmicutes_B: p = <0.0001, 0.0135, and <0.0001, for T80, PPI, and PPI + CAX, respectively).

T80 and PPI + CAX groups showed an increase in the relative abundance of Proteobacteria compared to the control group for female mice

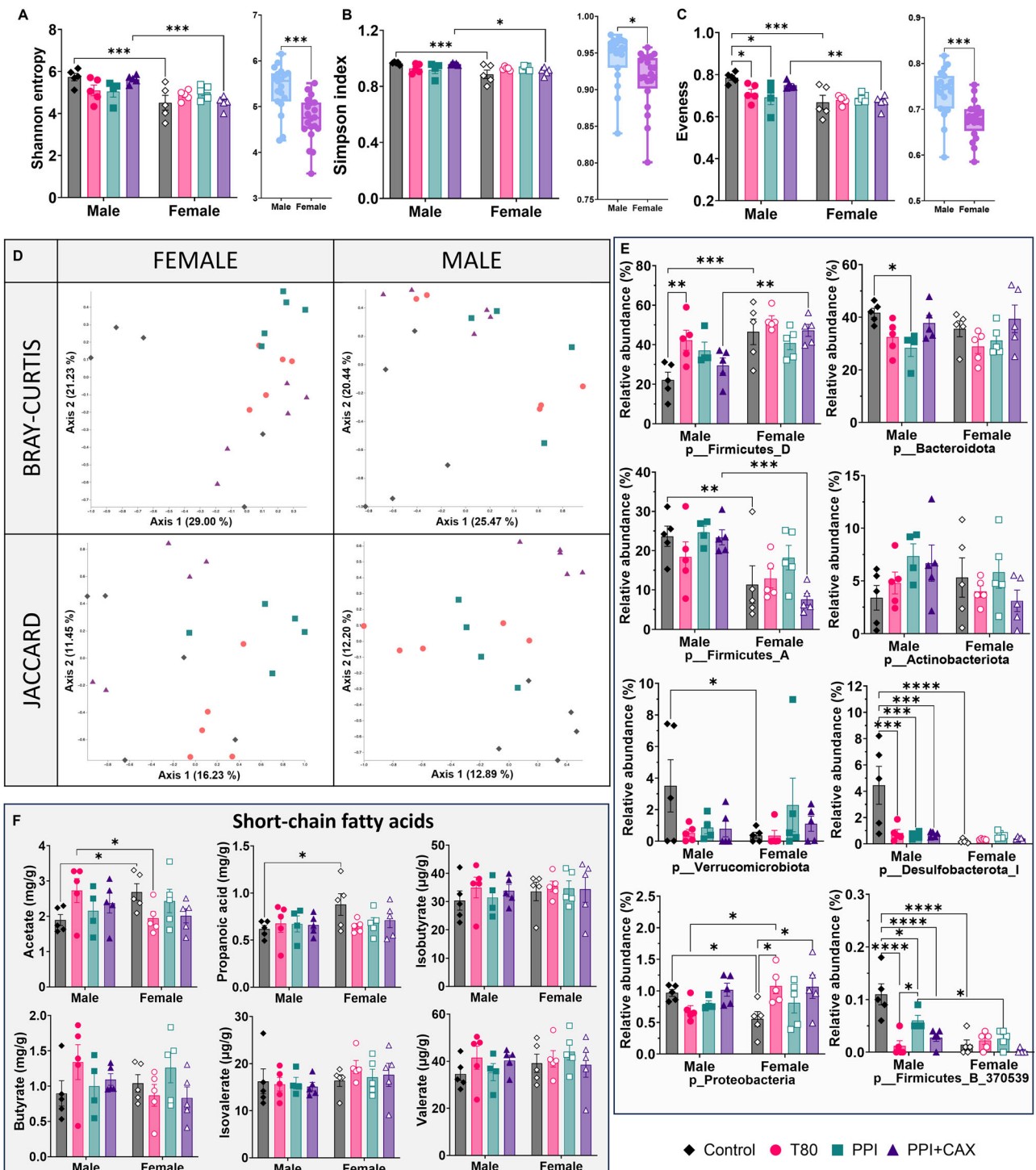

**Fig. 6 | Fecal microbiota and Short Chain Fatty Acids (SCFAs) analyses by sex after chronic exposure of mice to emulsifier-stabilized emulsions at dietary doses.** The olive-oil group (Control, diamonds), 2% Tween 80 group (T80, circles), 4% Pea protein isolate (PPI, squares), 4%PPI + 0.9% Corn arabinoxylans (PPI + CAX, triangles). Feces were collected in ARN-free tubes. Males are represented by full symbols and females by empty symbols. Fecal α-diversity **A** Shannon entropy, **B** Simpson index, and **C** Evenness, as well as β-diversity by sex **D** Bray- Curtis and Jaccard distances were analyzed at week 11. **E** Bar plot representation of the composition of fecal content at the phylum level. **F** Cecal concentrations of acetate, propanoic acid, isobutyrate, butyrate, isovalerate, and valerate. All graphic values are represented as mean ± SEM with n = 10 mice/group, including 5 males and 5 females. An asterisk indicates significant differences between groups (*p < 0.05; **p < 0.01; ***p < 0.001, ****p < 0.0001).

(p = 0.0195 and 0.238, respectively). A sexual dysmorphism was observed in the control group (p = 0.0201, lower relative abundance of Proteobacteria for females) and the T80 group (p = 0.0279, higher relative abundance of Proteobacteria for females).

Finally, emulsifiers had no effect on cecal SCFA content (Fig. 6H). A sex effect was observed in the control group regarding acetate and propanoic acid (higher values for females, p = 0.0391 and 0.164, respectively, Fig. 6H), and a sex effect was observed in the T80 group for acetate (lower value for

females, p = 0.0409) (Fig. 6H). For acetate, an interaction between sex and treatment was observed (p = 0.0257).

## Discussion

The aim of this study was to evaluate the potential of plant-based emulsifiers to replace conventional synthetic emulsifiers known to have a detrimental effect on health, at least when consumed at relatively high doses[14]. All the emulsions used in the study displayed similar physicochemical properties (see in ref. 26 for detailed information), which means that the effects observed are due to the emulsifiers used and not to the emulsions themselves.

In a first set of experiments, we showed that the combination of PPI and CAX allows to maintain vitamin $D_3$ bioaccessibility compared to T80, in agreement with previous data obtained for vitamin E and β-carotene[26,30]. The marked drop in vitamin $D_3$ bioaccessibility when using CAX alone was likely due to the fact that CAX is a dietary fiber neither digested nor absorbed in the small intestine[31]. Although the effect of fiber on vitamin D absorption is still unclear[32], it is possible that the presence of fiber slower both lipolysis and solubilization of lipolysis products in mixed micelles. The differential behavior of CAX when alone or in combination with legume proteins may be related to how emulsifiers distribute in the emulsion interface, which should be further explored. We then studied vitamin $D_3$ uptake by highly differentiated Caco-2 cell monolayers to determine the effect of emulsifiers on vitamin $D_3$ entry into the enterocytes. Decreased vitamin $D_3$ uptake by Caco-2 cells in the presence of PPI and CAX suggests that these emulsifiers, or other compounds associated with these emulsifiers, affect or slow down vitamin $D_3$ uptake, at least at the tested vitamin $D_3$ concentration of 0.23 μM. Vitamin $D_3$ uptake can take place either via membrane transporters (at concentrations below 2 μM) or by passive diffusion (at concentrations above 5–6 μM) (see for review[33] for detailed information of vitamin D transport across the enterocyte). We previously showed that PPI and CAX were associated with saponins and phytates (7.4 ± 1.15 and 5.4 ± 0.76 mg saponins/g of PPI or CAX, respectively; 24.8 ± 2.7 and 33.38 ± 4.48 mg phytates/g of PPI or CAX, respectively)[26]. As the presence of saponins, fiber and phytates can alter the uptake of vitamin K[34], it is therefore possible that saponins and phytates, together with the effect of CAX as a fiber, can also alter vitamin $D_3$ uptake in our experiments, maybe by affecting the vitamin $D_3$ transporter functioning.

While valuable for initial screening, in vitro results should be validated in vivo before application, as results may differ[23]. We thus explored the impact of these emulsifiers on lipid and vitamin $D_3$ postprandial responses and accumulation along the intestine in mice. For this set of experiments, 2 vitamin $D_3$ concentrations were chosen to ensure detection (216.54 and 2165.66 μM). Postprandial lipaemia can be modulated by modifying the structure and formulation of lipid-based foods[35]. In our study, both postprandial plasma TG response and TG accumulation in the intestine remained unchanged and similar to those previously observed[36], meaning that T80 can be successfully substituted by PPI + CAX without modifying postprandial lipaemia. Regarding vitamin $D_3$ responses, a ~15% decrease in absorption (AUC) was observed when vitamin $D_3$ was given at the lowest dose. This may be due to the presence of other compounds impairing/slowering down vitamin $D_3$ absorption via the impairment of transporters, as observed in Caco-2 cells. At high vitamin $D_3$ concentrations, vitamin $D_3$ postprandial responses were similar when using PPI + CAX and T80. This can be due to the fact that, at this pharmacological concentration, vitamin $D_3$ absorption is not mainly mediated by membrane transporters but mostly relies on passive diffusion[33]. Of note, in our second set of experiments, when mice were given emulsifiers at dietary doses 5 days a week for 11 weeks, the absorption of vitamins naturally present in the diet was not affected by emulsifiers. This confirms that at the tested doses, emulsifiers and their associated compounds did not impair vitamin transporters/mechanisms of absorption. These results are therefore promising for further applications in the food and pharmaceutical industries, at least when using high doses of vitamin $D_3$.

A major issue with the use of certain emulsifiers is their negative impact on gut health[14]. This effect, usually mediated by changes in colonic microbiota, has been well documented when emulsifiers were administrated in water at relatively high concentrations in rodents[7,9]. Chronic exposure to synthetic emulsifiers is known to impair the response of inflammatory biomarkers associated to plasma, feces or specific tissues, but up to date, no information on the impact of emulsifiers at dietary levels is available. We therefore investigated the potential inflammatory effect of PPI, PPI + CAX, and T80 at plausible daily exposure and incorporated into an oil-in-water emulsion (see Materials and Methods section and Supplementary Note 1), taking into consideration the sex of the animals. For this set of experiments, we decided not to include vitamin $D_3$ in the emulsions, because our aim was to specifically understand the effect of alternative emulsifiers on inflammation. Vitamin $D_3$ was previously shown to modulate intestinal microbiota, a regulation that could interact with the modulation driven by the emulsifiers. Moreover, because certain oils are known to disrupt microbiota and increase inflammation markers (e.g., medium-chain-triglyceride oil[37,38]), we chose to work with olive oil that does not display these negative effects[39]. Emulsions were administered to mice in a proportion equivalent to 10% of their food intake. This resulted in a T80 exposition of mice of 180 mg/kg/day, a dose 10-20 times lower than the doses used in previous studies (Chassaing et al.[9] or Viennois et al.[7]). This dose was equivalent to 14.59 mg/kg/day for humans[15], below the acceptable daily intake (25 mg/kg/day) proposed by the European Food Safety Authority[16]. Contrary to previous works (e.g., Chassaing et al.[9], Viennois et al.[7], or Liang et al.[40]), T80 was included in an emulsion in our study. This implied that the lipophilic tail of T80 (oleic acid) was integrated into the oil phase of the emulsion, which can limit microbiota exposition to the full structure of T80. For PPI, the exposition of mice was of 360 mg/kg/day, equivalent to 29.19 mg/kg/day for humans. For PPI + CAX the exposition of mice was of 360 mg PPI/kg/day and 81 mg CAX/kg/day, equivalent to 29.19 mg PPI/kg/day and 6.57 mg CAX/kg/day for humans[15]. For all groups, olive oil exposition was of 1 mL/kg/day.

At the tested levels of exposure, none of the emulsifiers, including T80, induced a marked inflammatory response, as shown by the overall unchanged cytokine and fecal biomarkers levels. However, the fact that LPS serum IgG levels were lower with PPI or PPI + CAX compared to T80 after 11 weeks of exposure, and that T80 showed a tendency towards higher levels than the control group, suggests that T80 regular consumption at a dietary dose could promote a significant increase in microbiota-derived antigen presentation to the underlying immune system. Lower LPS serum IgG response with PPI + CAX compared to the control condition from week 3 also suggests an anti-inflammatory effect of these emulsifiers. This hypothesis aligns with the observations of Li et al.[41], who reported a decrease of LPS serum levels after the inclusion of arabinoxylans in a high-fat diet in rats. Overall, these data showed that chronic exposure to PPI or PPI + CAX do not exacerbate the inflammatory response, but on the contrary, appeared to moderately improve it compared to T80 or olive oil alone.

Because PPI and PPI + CAX improved some inflammatory biomarkers, we speculated that these changes could be driven by morphological modifications and/or microbiota distribution in the colon and in the upper gastrointestinal tract, as previously observed[42,43]. The potential impact of emulsifiers on the upper gastrointestinal tract was investigated because: (i) recent research has highlighted that the microbiota found in the upper intestinal tract can play a significant role in the etiology of metabolic and non-infectious diseases[44,45] and emulsifiers can alter intestinal mucus structure[10], and (ii) it is the site of vitamin D absorption (see Fig. 1J, M). No effect of the emulsifiers was observed at the colon level (see Fig. S5). Daily exposure to dietary emulsifier concentrations for 11 weeks did not affect the morphology of the jejunum either. However, a different mucus layer phenotype was visually observed in the jejunum for T80 group, and T80 exposure favored bacterial proximity to the epithelium in male mice. In the small intestine, there is only one layer of mucus, which is thinner and looser than in the colon but enriched with antibacterial products, which limit the penetration of intestinal bacteria[46]. It has previously been shown that abnormal proximity of bacteria to the intestinal epithelium can be associated to changes in the mucus layer in both small intestine and colon[42,43,47] and that

exposure to T80 can increase the speed of transport in the mucus of certain bacteria, as *E. coli*[10]. In our study, jejunal goblet cells of male mice exposed to T80 seemed to be full of mucus compared to the other groups. Our observations were not confirmed when we analyzed genes related to mucus production and release in the small intestine. However, the increase in *Muc3* expression, along with the elevated expression of *Reg3-ɣ* observed in the proximal and median jejunum of female mice exposed to T80 compared to females exposed to PPI or PPI + CAX, may be attributed to enhanced mucus and antimicrobial peptide production to keep the intestinal microbiota at distance[48]. Bacteria reduced distance to the epithelium in T80 group could be linked with smoother bacterial transport within the mucus in the presence of T80, potentially via increased expression of genes associated with bacterial motility and flagellin production, as reported with higher concentrations of T80[10]. This reduced distance potentially suggests that even at low T80 exposure, expression of bacterial genes can be modified, although this hypothesis should be further confirmed. Reduced distance could also be favored by decreased local defenses and lower *IL6* in the median jejunum, allowing bacteria to approach the epithelium.

Finally, to assess if the tested emulsifiers at dietary doses could influence the fecal microbial community of the animals, as it happens with higher doses of emulsifiers[9], we analyzed the fecal microbiota composition at the end of the intervention period. Overall, the reduced bacteria distance to the epithelium observed in the small intestine, together with the higher fecal relative abundance of the phylum Firmicutes_D in males exposed to T80 compared to the control group, suggest that even at low concentration, chronic exposure to T80 can disbalance microbiota distribution and composition. Such effect was not observed with PPI or PPI + CAX emulsifiers. Our data are concordant with another study that reported increased Firmicutes relative abundance in the offspring microbial community after maternal T80 intake[40]. Increased abundance of Firmicutes has also been associated with the consumption of Western diets[49,50]. The significant sexual dimorphism observed supports the role of sex differences in shaping gut microbial communities in response to food intake[51]. The mechanism underlying the sex differences observed in this study was not directly investigated in the present work. However, the differences are probably linked to the sex chromosome genes and gonadal hormones that modulate the immune response[52]. These differences argue in favor of including both sexes in preclinical studies.

To conclude, our results show that PPI and CAX constitute possible alternatives to T80 for vitamin $D_3$ delivery. Our results also show that the negative effects of T80 on inflammation reported by other studies appears to be strongly linked to the grade of exposure (both concentration and chronicity), and potentially to co-exposure with other emulsifiers, as indicated by recent epidemiological studies[17,18,53]. In this study, we showed no evidence of systemic inflammation in healthy mice after exposure to PPI, PPI + CAX, or T80 at dietary doses. In the specific case of T80, a slight effect on (i) microbiota spatial distribution in the small intestine and (ii) fecal microbiota composition was observed. All these data support the potential of legume proteins and corn arabinoxylans as emulsifiers.

There are several limitations to this study. Firstly, the results of cellular vitamin $D_3$ uptake are specific to the Caco-2 TC7 cell line. While this model is widely used to assess vitamin uptake, other cell lines, such as LS174T or HT29 cells, could have provided additional information. When we addressed the impact of emulsifiers on mice health after chronic consumption, mice were on a control diet and considered healthy. Other studies suggested that food additives impact on biological responses can vary based on the subject health status, with unbalanced diets triggering negative effects less obvious in healthy individuals[54,55]. Therefore, further studies are needed to assess whether the effects of emulsifiers observed in this study could predispose mice to chronic pathologies, such as obesity, when exposed to unbalanced diets such as high-fat diets. Secondly, it would be useful to implement a longitudinal design to assess more precisely microbiota dynamics in response to such dietary intervention, in combination with the use of a host-free in vitro microbiota system. This would enable us to better elucidate the direct interactions between emulsifiers and gut microbiota.

Thirdly, our study revealed that emulsifiers altered the composition of the microbiota at the phylum level and promoted low-grade inflammation, but the underlying mechanisms—for instance which microorganisms are specifically involved—were not explored and are still unknown. An effect at the genus level was indeed observed (Fig. S10). Mechanisms by which emulsifiers induce changes in the intestinal microbiota, with the subsequent promotion of intestinal inflammation, are complex and multifactorial. Due to their broad-spectrum effects, emulsifiers appear to act on numerous members of the gut microbiota, notably increasing the abundance of Proteobacteria and other pro-inflammatory taxa[9], but also affecting gut microbiota transcriptional activity[8]. It was for example reported that emulsifier exposure led to increased expression of genes associated with bacterial motility and flagellin production, which can activate host innate immune responses via the Toll-like receptor 5 (TLR5) signaling pathways. This suggests that beyond merely shifting the microbiota structure, emulsifiers can also reprogram microbial gene expression in ways that promote chronic inflammation. Hence, emulsifiers likely act on multiple microbial taxa through both compositional and transcriptional mechanisms. Elucidating this point will require extended follow-up studies. Finally, results from murine models may not fully translate to humans due to biological differences. Although previous research shows that some harmful effects of emulsifiers at high doses reported in mice were also observed in humans[14], clinical data and epidemiological studies are needed to fully confirm our findings.

## Material and methods

### Material

Pea protein isolate (PPI; Nutralys®F85F, min 83% protein) was purchased from Roquette (Lestrem, France). Lentil protein isolate (LPI; LTVCP-80C, 80% protein) was kindly donated by AGT Foods (Regina, SK, Canada). Corn arabinoxylans (CAX; Soluble Fiber Corn, min 79.3% soluble fiber) were kindly donated by AgriFiber (Mundelein, IL, USA). All analytical grade chemicals were from Sigma-Aldrich (Saint-Quentin-Fallavier, France).

### Emulsion preparation

10% (w/w) oil-in-water emulsions were prepared by emulsifying olive oil and different PPI, LPI, and CAX aqueous solutions elaborated with Mili-Q water: 4% PPI (w/w), 4% LPI (w/w), 1% CAX (w/w), 4% PPI + 0.9% CAX (w/w), 4% PPI + 0.15% CAX (w/w), 4% LPI + 0.9% CAX (w/w), and 4% LPI + 0.15% CAX (w/w) solutions. Oil and aqueous solutions were pre-homogenized for 2 min at 12000 rpm with an Ultra-Turrax® (Ika, Staufen, Germany). Coarse emulsions were immediately homogenized with an Ultrasound Brandson 450 version 01.02 (Brookfield, Connecticut, USA) equipped with a 1/8″ diameter tapered probe at 70% amplitude during 6 min. During ultrasound homogenization, ice-cold ethanol was placed around the tubes to avoid an increase in temperature[26]. The detailed composition of emulsions is presented in Table 1. When loading the emulsions with vitamin $D_3$ (cholecalciferol), an appropriate volume of vitamin $D_3$ stock solution in HPLC grade ethanol was transferred to a glass tube, solvent was evaporated under nitrogen and the dried residue was solubilized in olive oil to achieve a concentration of 375 ng vitamin $D_3$/mL olive oil (emulsions for in vitro digestion), or either 8.33 or 0.83 mg vitamin $D_3$/mL olive oil (emulsions for postprandial experiments in mice). These concentrations were chosen to ensure an accurate detection in plasma. Samples were protected from light during all steps to avoid vitamin $D_3$ degradation. Final vitamin $D_3$ content of emulsions was 37.5 ng/mL (0.10 µM), 833 µg/mL (2165.66 µM), and 83.3 µg/mL (216.57 µM). Emulsions were used immediately.

### In vitro digestion of emulsions

The in vitro digestions of the different emulsions were carried out with a control meal. Briefly, 8 mL of 0.9% NaCl was added to a meal composed of 1.67 g of pureed potatoes and 0.3 g of pan-fried minced beef. The mixture was dispersed for 2 min with an Ultra-Turrax® (Ika) at 12000 rpm, before

1 g of the vitamin-loaded emulsion and 0.05 g of olive oil were added. A control emulsion was elaborated with an emulsifying solution of Tween 80 2% (w/w). For the control meal, water (1 g) and 0.15 g of extra virgin olive oil were added to match the same olive oil concentration as in the other samples. Next, 0.63 mL of artificial saliva (pH 7) was added to the mixture. The samples were incubated for 10 min at 37 °C in a shaking incubator. Subsequently, the pH was adjusted to 4.00 ± 0.02 with 1 M HCl and porcine pepsin (0.5 mL, 40 mg/mL in 0.1 M HCl) was added. The samples were incubated at 37 °C for 30 min to simulate the gastric step. The pH of the samples was then raised to 6.00 ± 0.02 with 0.9 M sodium bicarbonate (NaHCO$_3$), and 2.25 mL pancreatin (3 mg/mL in 0.1 M trisodium citrate pH 6) and 1 mL of porcine bile extract (127 mg/mL in 0.1 M trisodium citrate pH 6) was added. The samples were subsequently incubated in a shaking incubator at 37 °C for 30 min to simulate the duodenal step and to complete the digestion process. The final digesta was centrifuged (2000 × g for 1 h 12 min at 10 °C) and the supernatant was filtered with serial 0.8 μm and 0.2 μm filters (Millipore, Burlington, MA, USA)[26]. At all steps, samples were protected from light to avoid vitamin D$_3$ degradation. Aliquots of digesta (before centrifugation) and micelles (after centrifugation and filtration) were frozen at −80 °C until analysis. Each experiment was repeated in quadruplicate. Bioaccessibility was calculated as:

$$\text{Bioaccessibility}(\%) = \frac{\frac{\text{ng vitamin D3}}{\text{g micelles}}}{\frac{\text{ng vitamin D3}}{\text{g digesta}}} * \frac{\text{g micelles}}{\text{g digesta}} * 100$$

### Vitamin D$_3$ uptake in Caco-2 TC7 cell line

The Caco-2 TC7 cell line, a kind gift from Dr. M. Rousset (U178 INSERM, Paris, France), was routinely cultured in Dulbecco's Modified Eagle Medium (DMEM) containing 8% fetal bovine serum, 1% penicillin-streptomycin, and 1% non-essential amino acids at 37 °C in a 10% CO$_2$ atmosphere. Cells were grown on 12-well plates with inserts for 21 days before experiments[56]. Cytotoxicity of micellar fractions from in vitro digestions was assessed using different micelles concentrations (serial dilutions with serum-free complete medium up to 1/8). Cell viability was determined using an MTT assay as described elsewhere[57]. The dilution chosen was 1/8. To allow an accurate quantification, diluted micelles were enriched with vitamin D$_3$ to reach a concentration of 0.087 ng/μL (0.23 μM).

The day before the experiment, the media was replaced by a serum-free medium at both apical and basolateral sides. At the beginning of each experiment, cell monolayers (n = 4) were exposed to 1 mL of diluted micelles (apical side) and 2 mL of serum-free complete medium (basolateral side). Cells were then incubated for 4 h at 37 °C. Media and cells were collected in ice-cold PBS, and samples were stored at –80 °C until quantification of vitamin D$_3$. Vitamin D$_3$ absorption by the enterocytes was calculated as follows:

$$\text{Vitamin D3 uptake} = \frac{\text{vitamin D3 concentration in cells + basolateral media(if any)}}{\text{vitamin D3 concentration in diluted micelles}}$$

### In vivo experiments

The protocols were approved by the "Ministère de l'Education Nationale, de l'Enseignement Supérieur, et de la Recherche" (approval numbers APA-FIS#13473-2018020918403330v3 and APAFIS#46024-2023112013358971v4 for postprandial experiments and APAFIS#40225-202301031635597v4 for the chronic exposure experiment, respectively). We have complied with all relevant ethical regulations for animal use. 6-week-old C57BL/6 male and female mice were purchased from Janvier laboratory (Le Genest St Isle, France). Animals were housed in cages of 2-3 at a temperature-, humidity- and light-controlled room. They had free access to a standard chow diet (R03-25 irradiated, Safe, Augy, France) and tap water.

For postprandial experiments, experiments were conducted on 4 groups of 7-week-old male C57BL/6 J mice (20–30 g, n = 6 per group). Three days before the experiment, a blood sample was obtained at fast (zero baseline sample). Mice were fasted overnight and force-fed with 300 μL of vitamin D$_3$ enriched emulsions stabilized with either 2% T80 or 4% PPI + 0.9% CAX (PPI + CAX). Emulsions were enriched with 25 μg or 250 μg of vitamin D$_3$ (i.e., 1000 or 10000 IU). Blood samples were then taken at t = 1.5 h, 3 h, 4.5 h, and 6 h after gavage, collected in heparinized tubes, and immediately centrifuged (3000 g, 10 min 20 °C).

After euthanasia, liver and small intestine samples were harvested. Intestines were rinsed with ice-cold PBS and cut into 5–6-cm-long segments corresponding to duodenum, proximal jejunum, medium jejunum, distal jejunum, and ileum.

All samples were immediately snap-frozen in liquid nitrogen before storage at −80 °C until analysis.

For chronic exposure to emulsifiers, experiment was performed on 4 groups of 7-week-old male and female C57BL/6 J mice (20–30 g, 5 males and 5 females per group). Mice were exposed to olive oil (control group) or emulsions stabilized with 2% T80, 4% PPI, or 4% PPI + 0.9% CAX. For this study, we assumed as "plausible" a situation where an adult of 70 kg consumes 10% of his diet in the form of emulsion-based products (200 mL out of 2 kg of food) made with 10% of oil stabilized with a solution containing 2% T80, 4% PPI, or 4% PPI and 0.9% CAX (See Table S1 for detailed composition). These levels of emulsifiers are consistent with the levels used by the food industry[58]. For the mouse experiments, we substituted 10% of the diet with emulsion (200 μL out of 2–3 g of food for a 20 g mouse), resulting in an exposition of 180 mg/kg body weight/day for T80, 360 mg/kg body weight/day for PPI, and 441 mg/kg body weight/day for PPI + CAX. In this scenario, exposure to T80 when converted to human according to Reagan-Shaw et al.[15] is of 14.59 mg/kg body weight/day (see Supplementary Note 1 for detailed information about conversion). Values are below the acceptable daily intake (ADI) reported by the European Food Safety Agency (EFSA) (25 mg/kg body weight/day)[16]. For pea proteins and arabinoxylans, no ADI exists as these compounds can be present in our diet at much higher doses. Animals were force-fed with 200–300 μL of emulsions (10 μL/g mice) or 20 μL olive oil (control group) 5 days/week for 11 weeks. Force-feeding was carried out at 1:30 p.m. (± 30 min) every day, and emulsions were freshly prepared before each force-feeding. Food intake and body weight were measured weekly.

Fecal and blood samples of each individual were retrieved at weeks 0, 3, 6, 9, and 11. Blood samples were collected and centrifuged (3000 × g, 10 min, 20 °C). to recuperate the plasma. Feces were collected in ARN-free tubes. After euthanasia (week 11), the liver and spleen were weighed. Cecum was harvested. Small and large intestines were measured, and then 1 cm of jejunum, ileum, colon distal, and colon proximal samples were collected for histological analyses. The remaining small intestines were then rinsed with iced-cold PBS and cut into five segments (duodenum, proximal jejunum, median jejunum, distal jejunum, and ileum). Then, the mucosa of each segment was collected in ARN-free Eppendorf tubes for RT-qPCR analyses.

All samples were snap-frozen in liquid nitrogen and stored at −80 °C until analysis.

### Intestinal permeability assessment

Intestinal permeability was measured one day before euthanasia (week 11). Briefly, mice were fasted for 4 h, and force-fed with 100 μL (females) or 130 μL (males) of a solution of 4-kDa FITC-dextran (Sigma-Aldrich) at 80 mg/mL diluted in sterile PBS, to receive 0.4 mg 4-kDa FITC-dextran /g mice. Four hours after gavage, mice were anesthetized with isoflurane, blood was collected, and the plasma was collected as described above. Plasma fluorescence was immediately read in the microplate reader PerkinElmer EnSight™ (Waltham, MA, United States of America) with excitation of 485 nm and emission of 530 nm. A standard curve of 4-kDa FITC-dextran dissolved in plasma from ungavaged mice was used.

## Plasma biochemical analyses

Plasma cytokines IL-23, IL-1α, IFN-γ, TNF-α, MCP-1, IL-12p70, IL-1β, IL-10, IL-6, IL-27, IL-17A, IFN-β, and GM-CSF were measured at weeks 0, 3, 6, and 9 by flow cytometry with the kit LEGENDplex™ Mouse Inflammation Panel (13-plex) with V-bottom plate, following the instructions of the provider (BioLegend, Inc., San Diego, California, USA, Catalog number 740446). Quantification of plasma LPS and Flagellin FliC-specific plasma IgG were performed by ELISA. Microtitre plates were coated overnight with purified LPS (1 μg per well) or *E. coli* flagellin (100 ng per well) (Sigma). Diluted plasma samples (1:200) were then applied. After incubation and washing, wells were incubated with anti-mouse IgG Horseradish Peroxidase (CliniScience, 01017-05) for 1 h at 37 °C. Quantification was performed using the colorimetric peroxidase substrate tetramethylbenzidine (TMB), briefly, samples were incubated with TMB for 5 min in the dark at room temperature. Then, TMB stop solution was added and the optical density was read at 450 nm minus 540 nm. Data are reported as optical density corrected by subtracting background (determined by readings in plasma-free samples)[9].

## Fecal and cecal analyses

Fecal lipocalin 2 was measured according to Chassaing et al.[59]. Briefly, fecal samples were reconstituted in PBS containing 0.1% Tween 20 (100 mg/ml) and vortexed for 20 min to get a homogenous fecal suspension. These samples were then centrifuged for 10 min at 12,000 rpm and 4 °C. Clear supernatants were collected and stored at −20 °C until analysis. Lcn-2 levels were estimated in the supernatants using Duoset murine Lcn-2 ELISA kit (R&D Systems, Minneapolis, MN, Catalog number DY1857). Flagellin and LPS were measured as described in ref. 60 using HEK-Blue-mTLR5 and HEK-Blue-mTLR4 cells, respectively (Invivogen, San Diego, CA). We resuspended fecal material in PBS to a final concentration of 100 mg/ml and homogenized for 10 s using a Mini-Beadbeater-24 without the addition of beads to avoid bacteria disruption. We then centrifuged the samples at $8000 \times g$ for 2 min and serially diluted the resulting supernatant and applied to mammalian cells. Purified *E. coli* flagellin and LPS (Sigma, St.Louis, MO) were used as a positive control for HEK-Blue-mTLR5 and HEK-Blue-mTLR4 cells, respectively. After 24 h of stimulation, we applied cell culture supernatant to QUANTI-Blue medium (Invivogen, San Diego, CA) and measured alkaline phosphatase activity at 620 nm after 30 min. Acetate, propionate, butyrate, isobutyrate, valerate, and isovalerate from 35–115 mg of cecal content were converted to tert-butyldimethylsilyl derivatives and quantified by gas chromatography-mass spectrometry (GC-MS, Mass selective detector 5973 Network coupled to 6890 N GC Network GC System, Agilent Technologies, Santa Clara, California, USA) following the procedure described in ref. 42.

## 16S-based fecal microbiota composition analysis

**Bacterial DNA extraction.** DNA was extracted from frozen fecal samples using a QIAamp 96 PowerSoil Pro QIAcube HT kit (Qiagen Laboratories) with mechanical disruption (Qiagen TissueLyser II)[11]. Briefly, 650 μL of prewarmed buffer PW1 were added to fecal samples. Samples were thoroughly homogenized using bead-beating with a TissueLyser before centrifuging the plate at 4000 rpm for 5 min at 20 °C in order to pellet beads and particles. 400 μL of supernatant was added into a new 96 wells plate containing 150 μL of Buffer C3. After mixing and incubation on ice for 5 min, centrifugation was performed at 4000 rpm for 5 min at 20 °C. 300 μL of each supernatant were added to a new 96 well S-block plate, and 20 μL of Proteinase K were added and incubated for 10 min at room temperature. The following steps were next performed on a QIAcube high-throughput robot: addition of 500 μL of Buffer C4, DNA binding to a QIAamp 96 plate, column wash using AW1 (800 μL), AW2 (600 μL), and ethanol (400 μL), and DNA elution using ATE buffer (100 μL).

## Microbiota analysis by 16S rRNA gene sequencing using Illumina technology

16S rRNA gene amplification and sequencing were performed using the Illumina MiSeq technology[61,62]. The 16S rRNA genes, region V4, were amplified by PCR from each sample using a composite forward primer and a reverse primer containing a unique 12-base barcode, designed using the Golay error-correcting scheme, which was used to tag PCR products from respective samples[61]. The forward primer 515F was used: 5′-*AATGATAC GGCGACCACCGAGATCTACACGC*TXXXXXXXXXXXX**TATGGTAATT GT**GTGYCAGCMGCCGCGGTAA-3′: the italicized sequence is the 5′ Illumina adapter, the 12 X sequence is the Golay barcode, the bold sequence is the primer pad, the italicized and bold sequence is the primer linker, and the underlined sequence is the conserved bacterial primer 515F. The reverse primer 806 R used was 5′-*CAAGCAGAAGACGGCATACGAGAT***AGTCA GCCAGCC**GGACTACNVGGGTWTCTAAT-3′: the italicized sequence is the 3′ reverse complement sequence of Illumina adapter, the bold sequence is the primer pad, the italicized and bold sequence is the primer linker, and the underlined sequence is the conserved bacterial primer 806R. PCR reactions consisted of 5PRIME HotMasterMix (Quantabio, Beverly, MA, USA), 0.2 μM of each primer, 10–100 ng template, and reaction conditions were 3 min at 95 °C, followed by 30 cycles of 45 s at 95 °C, 60 s at 50 °C, and 90 s at 72 °C on a Biorad thermocycler. PCR products were visualized by gel electrophoresis. Products were then quantified (Quant-iT PicoGreen dsDNA assay), and a master DNA pool was generated from the purified products in equimolar ratios. The pooled products were purified with Ampure magnetic purification beads (Agencourt, Brea, CA, USA), quantified using the Quant-iT PicoGreen dsDNA assay, and sequenced using an Illumina MiSeq sequencer (paired-end reads, 2 × 250 bp) at the Genom'IC sequencing platform from Institut Cochin, Paris, France.

**16S rRNA gene sequences analysis.** 16S rRNA sequences were analyzed using QIIME2 – version 2022[63]. Sequences were demultiplexed and quality filtered using Dada2 method[64] with QIIME2 default parameters in order to detect and correct Illumina amplicon sequence data, and a table of Qiime 2 artifact was generated. A tree was next generated, using the align-to-tree-mafft-fasttree command, for phylogenetic diversity analyses, and α- and β-diversity analysis were computed using the core-metrics-phylogenetic command. For taxonomy analysis, features were assigned to operational taxonomic units (OTUs) with a 99% threshold of pairwise identity to the Greengenes reference database version 13.8[65].

## Intestinal tissue staining and analyses

One cm of jejunum, ileum, distal and proximal colon were collected in cold methanol-Carnoy's fixative solution (60% methanol, 30% chloroform, 10% glacial acetic acid) and incubated overnight at 4 °C. Samples were dehydrated, and embedded in paraffin according to the standard protocol for hematoxylin and eosin stain (HE) or immunofluorescence (IF) associated to the Fluorescence in situ hybridization (FISH), according to Hidalgo-Villeda et al.[42]. All the staining was done on dewaxed 8-μm sections. In IF experiments, antigen retrieval was performed in citric acid buffer 2 mM pH 6 for 45 min at 96 C and fluorescence-labeled secondary antibodies were used, and nuclei stained with DAPI. IF assay was performed with the anti-MUC2 antibody (1/500; sc-15334, Santa Cruz Biotechnology) and anti-VVA antibody (Vector Labs, B-1235-2) and the pan-bacteria probe Eub338-Alexa 555 5′-*GCTGCCTCCCGTAGGAGT-3″*. DNA was stained with Sytoxblue, revealing both eukaryotic and bacterial cells. Measures of villi length and crypt depth were performed in HE staining. At least 10 well U-shaped crypt-villus/animal were measured for 3–5 individual/sex*group with the SlideViewer 2.7 software. To measure the distance of the luminal bacteria to the epithelium, ZEISS ZEIN 3.7 software line tool was used (Carl Zeiss Microscopy).

## RT-qPCR

Total RNA was extracted from mucosa intestinal samples using a TRIzol reagent (Invitrogen™) according to the manufacturer's instructions. Then, 1 μg of total RNA was reverse transcribed into cDNA using M-MLV Reverse Transcriptase (Invitrogen™). Amplification of targeted genes was achieved using a Light Cycler® 480 (Roche Molecular Systems, Rotkreuz, Switzerland), and 18S rRNA was used as the endogenous control. Data were

analyzed using the LC480 software (Roche Diagnostics, Penzberg, Germany). The cDNA levels of *TNF-α, IL-6, CXCL-1, Reg3-γ, Muc2, Muc3, meprin-β*, and *Klf4* (list of used primers in Supplementary Table 1) were calculated for each sample using the cycle threshold (CT) and the Δ-CT method[66].

## Vitamin extraction and chromatographic analysis
### Vitamin A, D, and E extraction and quantification by HPLC.
Vitamins A and E from plasma (30 μL) from the chronic exposure experiment were extracted according to Reboul et al.[67] Dried extracts were dissolved in 50 μL of mobile phase (pure methanol) and 10–20 μL was used for HPLC analysis.

Vitamin $D_3$ from intestines and liver samples (100 mg intestine or liver/mL PBS) from the postprandial experiments were extracted from aqueous phases with the Bligh & Dyer method according to Goncalves et al.[68] Collected inferior phases were evaporated under nitrogen, and dried extracts were dissolved in 400 μL of mobile phase. A volume of 200 μL was used for HPLC analysis. The remaining volume was further dried, dissolved in 400 μL of isopropanol, and stored at −20 °C for posterior TG quantification.

Vitamin detection was performed by HPLC according to Antoine et al.[69] for vitamin $D_3$ and Reboul et al.[67] for vitamins A and E. Vitamins were identified by spectral analysis and retention time and co-injection in comparison with pure standards. Quantification was performed using Chromeleon 7.2 software (ThermoScientific, Villebon-sur-Yvette, France) to compare the peak area with standard reference curves.

### Vitamin D extraction and quantification by LC-MS/MS.
For in vitro sample extraction, 500 μL of sample (digesta, micelles, and cell samples) were added to a hemolysis tube. Then, 20 μL of a solution of deuterated vitamin $D_3$ (d3-vitamin $D_3$) in ethanol (40 ng/mL) were added as internal standard. 480 μL of ethanol solution were added to the sample, and the mixture was extracted twice with two volumes of hexane. The hexane upper phases obtained after centrifugation (1200 × g, 10 min, 4 °C) were collected with a glass pipette in a new hemolysis tube and evaporated under nitrogen.

For plasma extraction, 30 μL of plasma were added to a hemolysis tube. Then, 20 μL of a solution of d3-vitamin $D_3$ and d3-25(OH)$D_3$ in ethanol (40 ng/mL) were added as internal standards. 150 μL of acetonitrile were added. The mixture was vigorously mixed for 5 min and then centrifuged (1200 × g, 10 min, 4 °C), the supernatant was collected with a glass pipette in a new hemolysis tubs and evaporated under nitrogen.

The extracted vitamin $D_3$ and/or 25(OH)$D_3$ were derivatized with a solution of 4-phenyl-1,2,4-triazoline-3,5-dione (PTAD) in acetonitrile at a concentration of 4 mg/mL. PTAD solution (50 μL for in vitro samples or 25 μL for plasma) was added to the dry extract, and tubes were vigorously mixed for 10 min. This step was repeated twice. Ultrapure water was added to stop the reaction (20 μL for in vitro samples or 10 μL for plasma samples), and tubes were vigorously mixed for 5 min. The tubes were then evaporated under nitrogen.

All dry residues were dissolved in either or 100 μL (for in vitro samples) or 30 μL (for plasma samples) of acetonitrile and transferred to a vial for analysis by high-performance liquid chromatography-mass spectrometry/mass spectrometry (LC-MS/MS). The column was a Hypersil GOLD™ C18 column (Reference 25002-102130, ThermoFisher Scientific, Illkirch, France) maintained at a constant temperature (40 °C). The system was a TSQ Fortis Triple Stage Quadrupole MS/MS coupled to a Vanquish-Flex LC system (ThermoFischer Scientific). We used a heated ESI as ion source type with a pos ion spray voltage of 3500 V, ion transfer tube temperature of 300 °C, and vaporizer temperature of 350 °C. Chromeleon 7.2 Thermo-Fischer Scientific software was applied to set up, directly control, and process data. Mobile phase A consisted of Milli-Q water and formic acid (0.1%), and mobile phase B was made of acetonitrile and formic acid (0.1%). The flow was 0.4 mL/min and the following gradient was used: 0–1 min, 50% B; 1–13 min, linear gradient to 100% B; 13–15 min, 100% B; 15–16 min, linear gradient to 50% B; 16–20 min, 50% B. The injection volume was 5 μL. The

mass spectrometer was operated in a positive SRM scan mode and the following ion transitions were used: m/z 560.3847 → 298.1186 and m/z 563.4035 → 301.1374 for derivatized vitamin $D_3$ and derivatized d3-vitamin $D_3$, respectively, with a retention time of 11.37 min, and m/z 558.3690 → 298.1186 and m/z 561.3878 → 301.1374 for derivatized 25(OH)$D_3$ and derivatized d3-25(OH)$D_3$, respectively, with a retention time of 5.11 min. Vitamin $D_3$ and 25(OH)$D_3$ were quantified as the ratio of their peak area divided by the peak area obtained for the d3 solution, compared with standard reference curves.

## Triglyceride analysis
TG in plasma and intestine from postprandial experiments were measured with a kit for Total triglycerides-GPO Method from Biolabo (Les Hautes Rives, Maizy, France), following the instructions of the manufacturer. Measurements were directly performed in the plasma whereas for intestine and liver samples, TG were first extracted with the Bligh & Dyer method according to Goncalves et al.[68], as previously described in this work.

## Statistical analysis and reproducibility
Data were expressed as mean ± SEM. Statistical analyses were performed using GraphPad Prism software, version 10.2.3. (GraphPad Software, San Diego, California, USA) and SAS software version 9.4 (Cary, North Carolina, USA). All data were tested for normality before statistical analysis. The number of repetitions/animals was chosen to ensure a minimal number of repetitions for reliable statistical analysis. Group allocation was not blinded. For all experiments, the order of measurements was aleatory to maximize randomization. However, confounders were not controlled with a specific methodology. Data presented in Figs. 3, 5 and S4 were normalized compared with the control group and week 0, both defined as 1, to remove interferences of background inflammation and inflammation not related to the emulsifier treatment[70].

Bioaccessibility and cell uptake data were analyzed by one-way ANOVA. Tukey's test was used as a post-hoc test. Postprandial experiment data were analyzed using t-test (Area Under the Curve (AUC) of vitamin $D_3$ plasma concentration, liver samples) or two-way ANOVA (plasma and intestine samples). The interaction factor "time*mice group" (for plasma) or "part of the intestine*mice group" was used. to test whether the evolution of markers was significantly different between groups. Šídák's test was used as a post-hoc test, significant differences between groups were assessed for each time point/part of the intestine separately.

For FITC-Dextran intestinal permeability, organs weight and length, plasmatic vitamin content, fecal taxonomy relative abundance, short-chain fatty acid content, and RT-qPCR gene expression were analyzed using a two-way ANOVA + Tukey multiple comparison test.

Evolution of plasmatic cytokines, LPS, and Flagellin FliC-specific serum IgG, fecal inflammation markers, and comparisons between mice groups were analyzed using a mixed model. The interaction factor "time*mice group" was used to test whether the evolution of markers was significantly different between groups. The mixed models were based on a residual covariance structure called "compound symmetry" because it led to the lowest akaike information criterion (AIC). Normal distribution of residuals was verified.

The distance of bacteria to the epithelium and intestinal morphology were analyzed using a Nested 1-way ANOVA + Tukey multiple comparison test.

Values of p ≤ 0.05 were considered significant.

## Reporting summary
Further information on research design is available in the Nature Portfolio Reporting Summary linked to this article.

## Data availability
Data supporting this study are available at https://doi.org/10.57745/AFURVW[71]. Unprocessed sequencing data are deposited in the European Nucleotide Archive under accession number PRJEB89673.

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

## Acknowledgements

Ángela Bravo-Núñez postdoctoral contract "Margarita Salas" 2022-2024 was funded by the government of Spain (Ministerio de Universidades) and by the European Union (NextGeneration-EU). Ángela Bravo-Núñez post-doctoral contract 2025 was funded by the French Government ANR grant via Implanteus EUR (ANR-18-EURE-0009). Angélique Berthomé PhD fellow-ship was funded by Region Sud, France, and CTCPA Avignon (France). This work was partly funded by a Research Award from the French Nutrition Society attributed to Ángela Bravo-Núñez in 2022. Benoit Chassaing's laboratory is supported by a Starting Grant (grant agreement Invaders No. ERC-2018-StG-804135) and a Consolidator Grant (grant agreement Inter-Biome No. ERC-2024-CoG-101170920) from the European Research Council (ERC) under the European Union's Horizon 2020 research and innovation program, ANR grants EMULBIONT (ANR-21-CE15-0042-01) and DREAM (ANR-20-PAMR-0002). The authors would like to thank Jean-François Landrier (C2VN) for helpful discussions. The authors are also grateful to Flavie Sicard and Ljubica Svilar (CRIBIOM, Marseille, France), Teresa Gonzalez, Laura Beatrice Mattioli, and Vincent Bretegnier (C2VN, Marseille, France), Clara Delaroque and Héloïse Rytter (Institut Pasteur, Paris, France), Lionel Chasson and Clara Soufflet (CIML, Marseille, France), and CEFOS team (Marseille, France) for their support in the experiments. The authors warmly thank Matthieu Maillot (MS-Nutrition, Marseille, France) and Andrés Bravo-Núñez (Universidad de Valladolid, Spain) for helpful advice regarding statistical analysis. Cytokine assays were performed at AMUTI-CYT platform (C2VN, Marseille, France), short chain fatty acid analysis was performed at BIOMET platform (C2VN, Marseille, France). Imaging was performed at the PICSL imaging facility ImagImm (CIML, Marseille, France). 16s RNA analyses were performed at the Genom'IC sequencing platform (Institut Cochin, Paris, France).

## Author contributions

E.R. and A.B.N. conceived and designed the study. A.B.N. and J.T. performed the experiments with the inputs of A.B., C.S., D.V., J.C.M., K.A., and B.C. ABN analyzed the data. A.B.N. and E.R. wrote the original draft. J.T., B.C., and J.C.M. edited the manuscript. All authors commented and approved the final manuscript.

## Competing interests

The authors declare no competing interests.
