## [Transparent Peer Review file · Communications Biology]

Substitution of polysorbates by plant-based emulsifiers: impact on vitamin D bioavailability and gut health in mice

Corresponding Author: Dr Emmanuelle Reboul

This manuscript has been previously submitted at another journal. This document only contains information relating to versions considered at Communications Biology.

Version 0:

Reviewer comments:

Reviewer #1

(Remarks to the Author)

Comments to the Authors:

The manuscript entitled "Substitution of polysorbates by plant-based emulsifiers: impact on vitamin D bioavailability and gut health" focuses on a combination of pea protein and arabinoxylans appears as a sustainable alternative to polysorbates for vitamin D3 delivery. Honestly, it addresses an interesting point and talks. However, as reading, I feel disappointed. At current stage, I cannot agree to publish this work. I list my concerns in detail.

1. Your study revealed emulsifiers modify microbiota composition at the phylum level, but the underlying mechanism was still unknown, such as through what microorganism does the emulsifier cause intestinal inflammation.
2. The authors should increase clinical samples and epidemiological data to increase the persuasive power of the article.
3. The experimental design has flaws and the experimental methods are limited. For instance, in the in vitro experiment section, only one cell line was used, and in the in vivo experiment, only inflammatory markers were compared to verify that PPI+CAX improved intestinal inflammation compared to Tween 80 emulsion.
4. Why do you choose the dietary doses of emulsifiers in oil-in-water emulsions (180mg/kg/day, 5 days/week) for 11 weeks in the design of the experiment.
5. In animal experiments, it has always been found that gender has an impact on the results, but this aspect is not reflected in the titles or themes and the underlying mechanism is also not addressed.

Reviewer #2

(Remarks to the Author)

The current study explores the potential of replacing commonly used emulsifiers that suffer from detrimental impacts on the GI tract and microbiome (i.e. Tween 80) with naturally derived emulsifiers for the delivery of vitamin D. The concept is of interest, especially with the food industry seeking to move away from this potentially detrimental emulsifiers, towards safer emulsifiers. Overall, the study is of interest to a broad readership, but I do not necessarily believe Communications Biology to be the best fit, given this study is not strongly positioned within the field of biology. Rather, I think this manuscript would be better suited to a food science journal. I will, however, leave this to the editor to make the call. Despite my opinion on the suitability of the journal, I think this manuscript can be considered following major changes below:

1. The authors should highlight the important role that vitamin D plays in regulating the gut microbiome, given the aim for this study is to replace emulsifiers that disrupt the gut microbiome with 'safe' emulsifiers for vitamin D absorption.
2. Recent work has shown that oil-in-water emulsions in the form of self-emulsifying drug delivery systems (SEDDS) disrupt the gut microbiome in a composition dependent manner in rodents. Some of these systems contain T80, but other the findings suggest that a lipid phase is required for gut dysbiosis, rather than the emulsifier. This is interesting in the context of the current work because T80 performs similarly to the oil alone with respects to physiological characteristics and inflammatory biomarkers. This may be in fact due to the oil disrupting the microbiota to the same degree as the surfactant. The authors should consider discussing this in light of the recent research, see [dois: 10.1016/j.ijpharm.2023.123614](https://doi.org/10.1016/j.ijpharm.2023.123614) ; [10.1002/adfm.202403914](https://doi.org/10.1002/adfm.202403914) ; [10.1080/17425247.2023.2223937](https://doi.org/10.1080/17425247.2023.2223937)

3. How did the authors convert animal dosing to predicted human dose in line 60? Did this factor in the standard conversion based on FDA guidelines, or purely just a mass conversion based on relative weights of various species?
4. The introduction is lacking further detail as to the rationale behind selecting pea proteins and arabinoxylans. I assume these are already widely used and/or investigated as emulsifiers, and therefore this background is necessary.
5. What is the composition of the emulsion? It took me a long time to find this in the manuscript – these details should be upfront as it is critical information.
6. The results section needs improving. Given the methodology is presented last in this manuscript, it is important for the authors to at least briefly describe how data was measured/collected. E.g. how was bioaccessibility determined? What is this based on?
7. What were the physicochemical properties of the emulsions? Did these properties vary depending on the emulsifier used?
8. The authors should discuss the need for longitudinal microbiome sequencing to really identify changes to the gut microbiome based on interventions. It is very challenging to draw conclusions between various groups without the baseline for each animal.

Reviewer #3

(Remarks to the Author)

The manuscript by Bravo-Núñez and colleagues explores the potential of plant-based emulsifiers (PPI and CAX) as alternatives to T80, claiming that the combination of PPI and CAX results in comparable vitamin D3 bioavailability to T80 in vitro and in vivo in mice. The study further investigates the physiological and inflammatory responses to dietary doses of emulsifiers in oil-in-water emulsions, finding that T80, PPI, and PPI+CAX groups were similar to the control group (oil alone) in terms of physiological characteristics and inflammation biomarkers. Notably, LPS-specific serum IgG levels were reduced in the PPI and PPI+CAX groups compared to the T80 group. Additionally, exposure to T80, but not to PPI or PPI+CAX, was associated with reduced bacterial-epithelium distance in the jejunum and a significant increase in Firmicutes abundance. Based on these findings, the authors conclude that PPI and CAX could serve as viable alternatives to T80 for vitamin D3 delivery.

While PPI and CAX may contribute to vitamin D3 delivery, several scientific concerns must be addressed to support this conclusion. The study presents inconsistencies in vitamin D3 absorption data, an unclear rationale for dose selection, and a lack of mechanistic insights into inflammatory effects. Substantial revisions are needed to enhance the scientific rigor and clarity of the manuscript.

Review Comments:

Introduction

Line 55: In addition to citing a review, references to original research articles are warranted to support the discussion.

Line 58: While evaluating doses within the ADI range is valuable for real-world safety assessment, higher-dose toxicity testing is essential to define thresholds for adverse effects and clarify potential risks associated with chronic or accidental overconsumption exceeding ADI levels.

Line 130, Fig. 3: The study uses the following groups: 2% Tween 80 (T80), 4% PPI (PPI), and 4% PPI+0.9% CAX (PPI+CAX). What is the rationale for selecting these specific concentrations? How do they relate to physiological exposure levels?

Line 165: Given that previous studies have demonstrated significant emulsifier effects on colonic microbiota, it is crucial to verify whether PPI and CAX have any impact on the colon. Additionally, the authors should provide a clear scientific rationale for focusing on the jejunum rather than the colon.

Line 198, Fig. 5: While the study meticulously analyzes gene expression related to mucus production across three jejunal regions, these data do not seem fundamentally linked to the primary objective of evaluating PPI and CAX as vitamin D3 delivery agents.

Line 241, Fig. 6: The shift in focus to fecal microbiota, which is more relevant to colonic homeostasis, seems disconnected from the earlier sections of the study. The manuscript would benefit from better integration of these findings.

Line 249: The study lacks consistency in its vitamin D3 absorption findings. While PPI+CAX is claimed to maintain bioaccessibility similar to T80, Caco-2 cell experiments indicate reduced uptake, and in vivo data show a 15% decrease in absorption at low doses. Is this interpretation correct? These inconsistencies weaken the claim that PPI+CAX is a robust alternative to T80.

Line 450, Methods: How were the concentrations of T80, PPI, and CAX determined? A table outlining the corresponding dose per mouse would improve clarity and facilitate comparison with real-world exposure levels.

Version 1:

Reviewer comments:

Reviewer #1

(Remarks to the Author)

[Original language translated to English language via translate tool and curated by editor:

We thank the authors for their thoughtful response to the comments. The quality of the paper has been significantly improved

and most of the issues have been addressed. However, the following details still need further refinement. At this stage, I think this manuscript could consider the following changes. I listed my concerns in detail.

1. The author should add more clinical samples and epidemiological data to increase the persuasiveness of the article. We disagree that this should be a separate job.
2. The experimental design is flawed and the experimental methods are limited. For example, in the in vitro experiments section, only one cell line was used, you could have used 174 cells or HT29 cells or another cell line.]

Reviewer #3

(Remarks to the Author)

The manuscript by Bravo-Núñez and colleagues explores the potential of plant-based emulsifiers (PPI and CAX) as alternatives to T80, claiming that the combination of PPI and CAX results in comparable vitamin D₃ bioavailability to T80 in vitro and in vivo in mice. The study further investigates the physiological and inflammatory responses to dietary doses of emulsifiers in oil-in-water emulsions, finding that T80, PPI, and PPI+CAX groups were similar to the control group (oil alone) in terms of physiological characteristics and inflammation biomarkers. Notably, LPS-specific serum IgG levels were reduced in the PPI and PPI+CAX groups compared to the T80 group. Additionally, exposure to T80, but not to PPI or PPI+CAX, was associated with reduced bacterial-epithelium distance in the jejunum and a significant increase in Firmicutes abundance. Based on these findings, the authors conclude that PPI and CAX could serve as viable alternatives to T80 for vitamin D₃ delivery.

Most of the issues I pointed out - including the citation of original research, clarification of dosage rationale, and the scientific justification for focusing on the jejunum - have been somehow addressed through textual revisions, including additional data (e.g., Figure S5).

The authors have also explained the inconsistencies in vitamin D₃ bioavailability across different experimental models, pointing out that each model represents a distinct stage in the absorption process (bioaccessibility, cellular uptake, and in vivo absorption).

Overall, the author's responses have meaningfully improved the clarity and transparency of the study.

Point-by-point answers to reviewers' comments

Reviewer #1 (Remarks to the Author):

Comments to the Authors:

The manuscript entitled “Substitution of polysorbates by plant-based emulsifiers: impact on vitamin D bioavailability and gut health” focuses on a combination of pea protein and arabinoxylans appears as a sustainable alternative to polysorbates for vitamin D3 delivery. Honestly, it addresses an interesting point and talks.

We are grateful to the reviewer for this positive comment.

However, as reading, I feel disappointed. At current stage, I cannot agree to publish this work. I list my concerns in detail.

We are grateful for the feedback we received from the reviewer. We have tried to address any concerns, which was very useful in improving the quality of the work.

1. Your study revealed emulsifiers modify microbiota composition at the phylum level, but the underlying mechanism was still unknown, such as through what microorganism does the emulsifier cause intestinal inflammation.

Mechanisms by which emulsifiers induce changes in the intestinal microbiota, with the subsequent promotion of intestinal inflammation, are complex and multifactorial. As highlighted in previous work, emulsifiers such as carboxymethylcellulose (CMC) and Tween 80 (T80) were shown to drastically alter gut microbiota composition, particularly at the phylum level, notably increasing the abundance of Proteobacteria and other pro-inflammatory taxa (<https://doi.org/10.1038/nature14232>). Hence, pinpointing a single microorganism responsible for mediating inflammation is challenging due to the broad-spectrum effects of emulsifiers. Emulsifiers appear to act on numerous members of the gut microbiota, not only altering their composition but also affecting their transcriptional activity (<https://doi.org/10.1136/gutjnl-2016-313099>). It was for example reported that emulsifier exposure led to increased expression of genes associated with bacterial motility and flagellin production, which can activate host innate immune responses via TLR5 signaling pathways. This suggests that beyond merely shifting the microbiota structure, emulsifiers can also reprogram microbial gene expression in ways that promote chronic inflammation. Hence, with emulsifiers acting on multiple microbial taxa through both compositional and transcriptional mechanisms, answering this point will require extended follow-up studies.

We have modified the discussion to clarify this.

Changes can be found:

- Lines 391-396
- Lines 424-450

We have also included a general heatmap with the bacterial genus distribution of all mice (Figure S10).

Figure S10. Heatmap of individual fecal microbiota at the genus level after chronic exposure of mice to emulsifier-stabilized emulsions at dietary doses.

Feces were collected at week 11 in ARN-free tubes.

2.The authors should increase clinical samples and epidemiological data to increase the persuasive power of the article.

We believe that the article already gathers a fair amount of experiments/data. A clinical study would be of great interest and would thus constitute the next step to confirm our results. We believe that this should be a separate piece of work. Regarding epidemiological data, there are already studies connecting the consumption of synthetic emulsifiers and negative impact on health. These studies are cited in the manuscript:

- Line 68

At the end of the Discussion section, we have now included a statement making clearer the limitations of the present study, stating that our preclinical results should be validated in humans in the future. Changes can be found:

- Lines 424-450

3.The experimental design has flaws and the experimental methods are limited. For instance, in the *in vitro* experiment section, only one cell line was used, and in the *in vivo* experiment, only inflammatory markers were compared to verify that PPI+CAX improved intestinal inflammation compared to Tween 80 emulsion.

The reviewer is right, in the *in vitro* experiment section, only Caco-2 cells were used. The reason behind this choice is that the Caco-2 cell line is a well-established and recognized cell model, that has been widely used to characterize fat-soluble vitamin uptake, including vitamin D. The relevance of this line is well evidenced in the following review <https://doi.org/10.1016/j.plipres.2022.101208>.

Regarding the *in vivo* experiments, we selected several inflammatory markers (intestinal permeability, plasma cytokines IL-23, IL-1 α , IFN- γ , TNF- α , MCP-1, IL-12p70, IL-1 β , IL-10, IL-6, IL-27, IL-17A, IFN- β , and GM-CSF, as well as plasma LPS and Flagellin FliC-specific plasma IgG, fecal lipocalin 2, fecal flagellin, fecal LPS, cDNA levels of intestinal genes) because our interest was to explore the potential inflammatory effect of the emulsifiers. In addition to inflammatory markers, we also examined the genes involved in mucus production and secretion, a crucial aspect of intestinal defense that is widely studied in chronic and acute inflammation. These studies were performed using both qPCR and immunostaining. In addition, analyses of SCFAs, whose concentrations can reveal dysbiosis and indicate a pro- or anti-inflammatory character, were performed using state-of-the-art technologies. We are not sure to understand the point the reviewer.

4.Why do you choose the dietary doses of emulsifiers in oil-in-water emulsions (180mg/kg/day, 5 days/week) for 11 weeks in the design of the experiment.

These doses were selected because they represent plausible dietary doses, as initially justified in the Supplementary Information (we believe reviewers did not have access to this file during the first round of reviewing as it's a recurrent remark). Other reviewers asked the same question, and therefore, we have now included this information in the main text. We have chosen 11 weeks because, taking into consideration that the lifespan of mice is of 2 years, 11 weeks of exposure would be equivalent to ~10 years of consumption in humans with a lifespan of 80 years. Changes to clarify these concerns can be found:

- Lines 128-133
- Lines 351-354
- Lines 550-563
- Supplementary information.

5.In animal experiments, it has always been found that gender has an impact on the results, but this aspect is not reflected in the titles or themes and the underlying mechanism is also not addressed.

We have now included this in the abstract of the work. In the discussion, we have also highlighted that both sex were used, notably:

- Line 337

Finally, although the mechanisms were not specifically addressed, we have included a statement addressing the possible underlying mechanism for sex differences, that can be found:

- Lines 409-413.

Reviewer #2 (Remarks to the Author):

The current study explores the potential of replacing commonly used emulsifiers that suffer from detrimental impacts on the GI tract and microbiome (i.e. Tween 80) with naturally derived emulsifiers for the delivery of vitamin D. The concept is of interest, especially with the food industry seeking to move away from this potentially detrimental emulsifiers, towards safer emulsifiers. Overall, the study is of interest to a broad readership, but I do not necessarily believe Communications Biology to be the best fit, given this study is not strongly positioned within the field of biology. Rather, I think this manuscript would be better suited to a food science journal. I will, however, leave this to the editor to make the call. Despite my opinion on the suitability of the journal, I think this manuscript can be considered following major changes below.

We are grateful to the reviewer for pointing out that the study is of interest to a broad readership. We are also thankful for her/his remarks they had a positive impact on the improvement of the present work.

1. The authors should highlight the important role that vitamin D plays in regulating the gut microbiome, given the aim for this study is to replace emulsifiers that disrupt the gut microbiome with 'safe' emulsifiers for vitamin D absorption.

We agree with the reviewer that vitamin D plays a role in regulating the gut microbiome, but this regulation was outside the scope of the present work. We sought to identify emulsifiers capable of efficiently delivering vitamin D without inducing inflammation.

The data from this work actually goes beyond vitamin D delivery, as emulsifiers of plant origin can also be used in the food or pharmaceutical industries for other applications.

Because vitamin D is known to regulate gut microbiome and also to decrease inflammation, we decided not to include vitamin D in the emulsions for chronic exposure. Our aim was to evaluate the effect of emulsifiers alone. This is now clarified in the text to avoid confusion:

- Lines 337-343

We believe that the interaction between emulsifiers-vitamin D is also of great interest, and this will be further explored in the future.

2. Recent work has shown that oil-in-water emulsions in the form of self-emulsifying drug delivery systems (SEDDS) disrupt the gut microbiome in a composition dependent manner in rodents. Some of these systems contain T80, but other the findings suggest that a lipid phase is required for gut dysbiosis, rather than the emulsifier. This is interesting in the context of the current work because T80 performs similarly to the oil alone with respects to physiological characteristics and inflammatory biomarkers. This may be in fact due to the oil disrupting the microbiota to the same degree as the surfactant. The authors should consider discussing this in light of the recent research, see DOIs: 10.1016/j.ijpharm.2023.123614 ; 10.1002/adfm.202403914 ; 10.1080/17425247.2023.2223937.

In the present study, we have used olive oil, while the works suggested by the reviewer used MCT oil. MCT oil is known to aggravate intestinal inflammation (<https://doi.org/10.1016/j.intimp.2016.03.019>). In fact, intestinal inflammation observed with MCT was not observed when using olive oil.

In addition, several studies support olive oil not having inflammatory effects (see for review <https://doi.org/10.1016/j.tifs.2018.05.001>), therefore, the hypothesis that our lipid phase (olive oil) can potentially have a negative effect in an equal manner than Tween 80 via disruption of the microbiota is unlikely. In our opinion, the lack of differences between the control and T80 groups is due to the fact that T80 does not induce systemic inflammation at low doses in healthy individuals, as stated in the final conclusion of the present work, in:

- Lines 415-423

We have also clarified in the Discussion the choice of olive oil and cited 2 of the suggested references:

- Lines 341-343

3. How did the authors convert animal dosing to predicted human dose in line 60? Did this factor in the standard conversion based on FDA guidelines, or purely just a mass conversion based on relative weights of various species?

The dose was converted according to the indications of the cited article (Reagan - Shaw, S., Nihal, M. & Ahmad, N. Dose translation from animal to human studies revisited. *FASEB j.* 22, 659 – 661 (2008). <https://doi.org/10.1096/fj.07-9574lsf>), using their equation. The parameters of conversion of the equation take into consideration several parameters of biology, including oxygen utilization, caloric expenditure, basal metabolism, blood volume, circulating plasma proteins, and renal function. The exact equation is presented in the Supplementary Material.

4. The introduction is lacking further detail as to the rationale behind selecting pea proteins and arabinoxylans. I assume these are already widely used and/or investigated as emulsifiers, and therefore this background is necessary.

We thank the reviewer for the comment. In addition to the already cited literature, we have included 2 review articles that support the effectiveness of legume proteins and arabinoxylans as emulsifiers in the Introduction section:

- Line 74.

5. What is the composition of the emulsion? It took me a long time to find this in the manuscript – these details should be upfront as it is critical information.

We apologize for this; we believe that the supplementary material was not made available to the reviewers. We have included a Table (Table 1) with the detailed composition of all emulsions.

Table 1. Detailed emulsion formulation (mg/g emulsion)

	Tween 80	Pea protein isolate	Lentil protein isolate	Corn arabinoxylans	MiliQ Water	Olive oil*
T80	18	-	-	-	882	100
PPI	-	36	-	-	864	100

PPI+0.9% CAX**	-	36	-	8.1	855.9	100
PPI+0.15%CAX	-	36	-	1.35	865.65	100
LPI	-		36	-	864	100
LPI+0.9% CAX	-	-	36	8.1	855.9	100
LPI+0.15%CAX	-	-	36	1.35	862.65	100

*Oil was loaded with vitamin D for bioaccessibility and bioavailability experiments at low and high doses to achieve the vitamin D concentration in the emulsions of 0.975 and 216.54 or 2165.66 μM , respectively. For cell uptake, diluted micelles (1/8) from bioaccessibility experiments were spiked with vitamin D to achieve a final concentration of 0.23 μM . **Also referred as PPI+CAX in the text.

6. The results section needs improving. Given the methodology is presented last in this manuscript, it is important for the authors to at least briefly describe how data was measured/collected. E.g. how was bioaccessibility determined? What is this based on?

We have revised the results section. Although we have tried to make it more understandable, we tried to keep the results section as factual as possible, avoiding detailed explanations that could potentially overlap with the Discussion. The story of what led us to do the analysis reported is described in the Discussion section.

6. What were the physicochemical properties of the emulsions? Did these properties vary depending on the emulsifier used?

Physicochemical properties of the emulsions were reported elsewhere (see 10.1016/j.foodchem.2024.139820). This is now clarified in the Introduction and in the Discussion:

- Lines 75-78
- Lines 279-283

8. The authors should discuss the need for longitudinal microbiome sequencing to really identify changes to the gut microbiome based on interventions. It is very challenging to draw conclusions between various groups without the baseline for each animal.

We appreciate the reviewer's insightful comment. We fully agree that longitudinal microbiome sequencing is essential to accurately assess microbiota dynamics in response to dietary interventions, especially given the inter-individual variability and the importance of baseline measurements. In future studies, it would be relevant to implement such longitudinal designs, in combination with the use of a host-free *in vitro* microbiota system, in order to better elucidate the direct interactions between emulsifiers and the intestinal microbiota. This has been stated in the limitations of the present study.

Reviewer #3 (Remarks to the Author):

The manuscript by Bravo-Núñez and colleagues explores the potential of plant-based emulsifiers (PPI and CAX) as alternatives to T80, claiming that the combination of PPI and CAX results in comparable vitamin D3 bioavailability to T80 *in vitro* and *in vivo* in mice. The study further investigates the physiological and inflammatory responses to dietary doses of emulsifiers in oil-in-water emulsions, finding that T80, PPI, and PPI+CAX groups were similar to the control group (oil alone) in terms of physiological characteristics and inflammation biomarkers. Notably, LPS-specific serum IgG levels were reduced in the PPI and PPI+CAX groups

compared to the T80 group. Additionally, exposure to T80, but not to PPI or PPI+CAX, was associated with reduced bacterial-epithelium distance in the jejunum and a significant increase in Firmicutes abundance. Based on these findings, the authors conclude that PPI and CAX could serve as viable alternatives to T80 for vitamin D3 delivery.

While PPI and CAX may contribute to vitamin D3 delivery, several scientific concerns must be addressed to support this conclusion. The study presents inconsistencies in vitamin D3 absorption data, an unclear rationale for dose selection, and a lack of mechanistic insights into inflammatory effects. Substantial revisions are needed to enhance the scientific rigor and clarity of the manuscript.

We are grateful to the reviewer for the provided feedback that helped us to improve the present work.

Review Comments:

Introduction

Line 55: In addition to citing a review, references to original research articles are warranted to support the discussion.

Original research articles are now cited in:

- Line 55.

Line 58: While evaluating doses within the ADI range is valuable for real-world safety assessment, higher-dose toxicity testing is essential to define thresholds for adverse effects and clarify potential risks associated with chronic or accidental overconsumption exceeding ADI levels.

We agree with the observation of the reviewer; we never stated that we found that the cited studies are not valuable. We believe they are, and in fact, one of the coauthors of the present manuscript was involved in many of these studies. The aim of this work was to complement the available information with a study using lower doses. We rephrased the sentence to avoid confusion. Changes have been made in:

- Lines 55-62

Line 130, Fig. 3: The study uses the following groups: 2% Tween 80 (T80), 4% PPI (PPI), and 4% PPI+0.9% CAX (PPI+CAX). What is the rationale for selecting these specific concentrations? How do they relate to physiological exposure levels?

These specific concentrations were chosen because they correspond to the levels required to obtain a stable emulsion over time. (see our previous publication with the same emulsifiers <https://doi.org/10.1016/j.foodchem.2024.139820>)

For pea proteins and arabinoxylans, the physiological exposure levels can be largely above the levels used in our emulsions, as plant proteins and fiber can be consumed in diets at much higher doses.

For Tween80, exposure is below the ADI proposed by the European Safety authorities. We believe reviewers didn't had access to supplementary materials where this was explained. We have now updated the Supplementary Material and included this information in the main text. Changes have been made in:

- Lines 351-354
- Lines 550-563

Line 165: Given that previous studies have demonstrated significant emulsifier effects on colonic microbiota, it is crucial to verify whether PPI and CAX have any impact on the colon. Additionally, the authors should provide a clear scientific rationale for focusing on the jejunum rather than the colon.

In fact, no impact was observed on the colon, this is why we did not include the images in the first version of the manuscript. The images supporting these observations are now included in Figure S5.

Figure S5. Colonic characteristics after chronic exposure of mice to emulsifier-stabilized emulsions at dietary doses.

Representative spectral confocal imaging projections of median jejunum from W11 mice stained by Fluorescence *in situ* Hybridization (FISH) for all bacteria (Eub-338 probe, orange), MUC2 (magenta) and Vicia Villosa Agglutinin (VVA, cyan). Mice were euthanized at fast. Bars, 50 μm separated by sex.

The scientific rationale for focusing on the jejunum is that very recent data highlighted that the microbiota found in the upper intestinal tract can play a significant role in the etiology of metabolic and non-infectious diseases (see <https://doi.org/10.1016/j.metabol.2023.155712> and <https://doi.org/10.1111/mmi.15270>). In addition, the jejunum plays a key role in the absorption of micronutrients, including vitamin D. We have modified the discussion to make this point clear to the reader. Changes can be found in:

- Lines 370-375

Line 198, Fig. 5: While the study meticulously analyzes gene expression related to mucus production across three jejunal regions, these data do not seem fundamentally linked to the primary objective of evaluating PPI and CAX as vitamin D3 delivery agents.

The interest of analyzing gene expression in the jejunal regions is linked to our observation that bacteria were closer to the intestinal cells after exposure to specific emulsifiers in these specific areas. We thus evaluated whether this proximity could be due to a difference in terms of mucus secretion, and whether this could lead to inflammation. We have clarified this in the Discussion section in:

- Lines 209-212
- Lines 233-234

Line 241, Fig. 6: The shift in focus to fecal microbiota, which is more relevant to colonic homeostasis, seems disconnected from the earlier sections of the study. The manuscript would benefit from better integration of these findings.

We have tried to better integrate the reasons that took us to look into fecal microbiota:

- Lines 398-400

Line 249: The study lacks consistency in its vitamin D3 absorption findings. While PPI+CAX is claimed to maintain bioaccessibility similar to T80, Caco-2 cell experiments indicate reduced uptake, and in vivo data show a 15% decrease in absorption at low doses. Is this interpretation correct? These inconsistencies weaken the claim that PPI+CAX is a robust alternative to T80.

Yes this is correct. However, each experiment explores a different step of the absorption process, so data are not inconsistent. Bioaccessibility reflects what happens at the end of the digestion process, Caco2 cell experiment evaluates the uptake of vitamin D, while mouse experiment gives an overview of the whole digestion-absorption process.

Line 450, Methods: How were the concentrations of T80, PPI, and CAX determined? A table outlining the corresponding dose per mouse would improve clarity and facilitate comparison with real-world exposure levels.

A table with the specific composition of the emulsions is now included in the manuscript (Table 1). The doses of T80, PPI or PPI+CAX consumed daily by mice are included:

- Lines 351-354
- Lines 550-563

Table 1. Detailed emulsion formulation (mg/g emulsion)

	Tween 80	Pea protein isolate	Lentil protein isolate	Corn arabinoxylans	MiliQ Water	Olive oil*
T80	18	-	-	-	882	100
PPI	-	36	-	-	864	100
PPI+0.9% CAX**	-	36	-	8.1	855.9	100
PPI+0.15%CAX	-	36	-	1.35	865.65	100
LPI	-	-	36	-	864	100
LPI+0.9% CAX	-	-	36	8.1	855.9	100
LPI+0.15%CAX	-	-	36	1.35	862.65	100

*Oil was loaded with vitamin D for bioaccessibility and bioavailability experiments at low and high doses to achieve the vitamin D concentration in the emulsions of 0.975 and 216.54 or 2165.66 μM , respectively. For cell uptake, diluted micelles (1/8) from bioaccessibility experiments were spiked with vitamin D to achieve a final concentration of 0.23 μM . **Also referred as PPI+CAX in the text.